# PDE-Transformer: Efficient and Versatile Transformers for Physics Simulations

Benjamin Holzschuh [1]   Qiang Liu [1]   Georg Kohl [1]   Nils Thuerey [1]

## Abstract

We introduce PDE-Transformer, an improved transformer-based architecture for surrogate modeling of physics simulations on regular grids. We combine recent architectural improvements of diffusion transformers with adjustments specific for large-scale simulations to yield a more scalable and versatile general-purpose transformer architecture, which can be used as the backbone for building large-scale foundation models in physical sciences. We demonstrate that our proposed architecture outperforms state-of-the-art transformer architectures for computer vision on a large dataset of 16 different types of PDEs. We propose to embed different physical channels individually as spatio-temporal tokens, which interact via channel-wise self-attention. This helps to maintain a consistent information density of tokens when learning multiple types of PDEs simultaneously. We demonstrate that our pre-trained models achieve improved performance on several challenging downstream tasks compared to training from scratch and also beat other foundation model architectures for physics simulations. Our source code is available at `https://github.com/tum-pbs/pde-transformer`.

## 1. Introduction

Large multi-purpose networks that are trained on diverse, high-quality datasets have shown great achievements when adapted to specific downstream tasks. These so-called *foundation models* have received widespread recognition in fields such as computer vision (Yuan et al., 2021; Awais et al., 2025), decision making (Yang et al., 2023b), and time series prediction (Liang et al., 2024). Naturally, these models are likewise extremely interesting for the scientific machine learning community (Bodnar et al., 2024; Herde et al., 2024; Zhang et al., 2024a), where they can be used in a variety of downstream tasks such as surrogate modeling (Kim et al., 2019; Sun et al., 2020) or inverse problems involving physics simulations (Ren et al., 2020; Holzschuh et al., 2023). They are especially promising in areas where only low amounts of training data are available.

The adoption of machine learning for physics simulations faces several characteristic challenges. The underlying physics often exhibit an inherent multi-scale nature (Smith, 1985). The representation of data is tied to the numerical methods used for simulations, ranging from regular grids to meshes to particle-based representations (Anderson et al., 2020). While numerical simulations produce vast amounts of data, they oftentimes cannot readily be used as input to machine learning models as further preprocessing is required. Additionally, machine learning models in this area directly compete with traditional numerical methods. As such, they need to either outperform such solvers in terms of accuracy or speed while exhibiting high reliability. Alternatively, they need to provide solutions that solvers cannot easily obtain, for example, uncertainty estimates (Geneva & Zabaras, 2019; Jacobsen et al., 2025; Liu & Thuerey, 2024; Kohl et al., 2024) or working with partial inputs (Wu et al., 2024a).

We address these problems by introducing a multi-purpose transformer architecture *PDE-Transformer* that is carefully tailored to scientific machine learning: the same architecture can be used to generalize between different types of PDEs, different resolutions, domain extents, boundary conditions, and it includes deep conditioning mechanisms for PDE- and task-specific information. Our model works on regular grids in two dimensions, and can be trained in a supervised manner as an efficient surrogate model, or as a diffusion model for downstream tasks where solutions have wider posterior distributions. Specifically, our contributions are:

- We augment existing SOTA diffusion transformer architectures to tailor them to PDE and physics simulations, among others, by token down- and upsampling for efficient multi-scale modeling and shifted window attention for improved scaling to high-resolution data.

- We modify the attention operation to decouple token interactions between the spatio-temporal axes and the physical channel axis. This improves accuracy and

[1] School of Computation, Information and Technology, Technical University of Munich, Germany. Correspondence to: Benjamin Holzschuh <benjamin.holzschuh@tum.de>.

*Proceedings of the 42nd International Conference on Machine Learning*, Vancouver, Canada. PMLR 267, 2025. Copyright 2025 by the author(s).

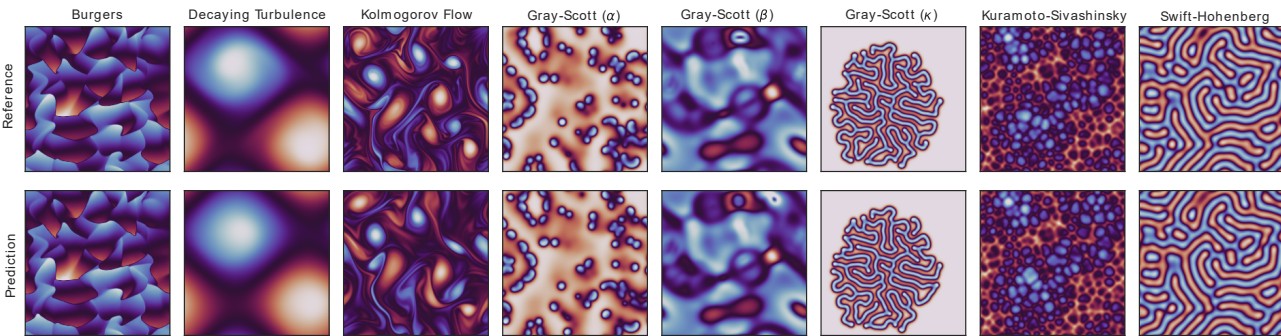

*Figure 1.* PDE-Transformer is a transformer model tailored to scientific data, the images above show its autoregressive predictions after 20 time steps on a large dataset comprising 16 different PDE dynamics, given initial conditions only. Additional simulation parameters (viscosity, domain extent, etc.) are unknown to the model and need to be inferred from the observed data. PDE-Transformer is especially well suited to be pre-trained for out-of-distribution downstream tasks.

allows for better generalization to different PDEs.

- We perform a detailed ablation study on accuracy-compute tradeoffs when scaling and modifying PDE-Transformer.

- We demonstrate that PDE-Transformer generalizes well to challenging downstream tasks when pre-trained on a set of generic PDEs.

## 2. Related Work

**Transformers** Transformers have become one of the dominant deep learning architectures. While first established for sequence-to-sequence models in natural language processing (Vaswani et al., 2017), transformers have been successfully applied to computer vision by splitting images into patches, which are embedded jointly with their position as tokens (Dosovitskiy et al., 2021, ViT). ViTs have been shown to scale better in vision tasks than classical convolutional neural networks and benefit from high accelerator utilization (Maurício et al., 2023; Takahashi et al., 2024; Rodrigo et al., 2024). Large transformer-based architectures have led to *foundation models* such as GPT (Radford et al., 2018; 2019; Brown et al., 2020) and BERT (Devlin et al., 2019).

**Diffusion Transformers** While early applications of transformers in computer vision focus on visual recognition tasks, the success of latent diffusion models has also led to the adoption of transformer-based diffusion models, the diffusion transformer (Peebles & Xie, 2023, DiT). Instead of modeling the data space directly, latent diffusion models (Ho et al., 2020; Rombach et al., 2022) use a variational autoencoder to embed the data and operate in the resulting latent space. Additionally, diffusion transformers have very powerful conditioning mechanisms based on adaptive layer normalization (Perez et al., 2018) in which

class labels or text encodings can be used as an additional input. Diffusion transformers use a significantly lower patch size than many previous transformer models for computer vision tasks. While diffusion models show an increasing performance with lower patch sizes, they can only be used for high-dimensional data by working in the latent space. This is caused by the quadratic scaling of the required compute with the number of tokens, due to the global self-attention mechanisms of the diffusion transformer architecture, making training and inference computationally expensive. In this paper, we focus on modeling the raw data directly without including any pre-trained autoencoders.

**Attention mechanisms** The global self-attention operation of transformers is the main computational bottleneck of the transformer architecture. There have been two main directions addressing this. Windowed attention limits the computation of self-attention to non-overlapping local windows (Liu et al., 2021). For different layers, this window is shifted to prevent discontinuities at the window borders. Simiarly, in axial attention (Ho et al., 2019), tokens only interact with each other along a specific axis, i.e. when they belong to the same image row or column. When combined with down- and upsampling in a hierarchical architecture, the resulting models show the same or improved performance while operating at a lower compute cost. The cost of attention can also be lowered by algorithmic modifications to the attention mechanisms, which make the attention scale linearly (Wang et al., 2020; Cao, 2021). However, these models often report subpar performance compared to the quadratic self-attention. For physics simulations, the importance of transformers and attention mechanism has likewise been recognized. Galerkin transformers (Cao, 2021) utilize a linearized attention variant that removes the softmax normalization and can be related to a learnable layer-wise Petrov-Galerkin projection. Multiple Physics Pertaining (McCabe et al., 2023, MPP) uses a custom transformer

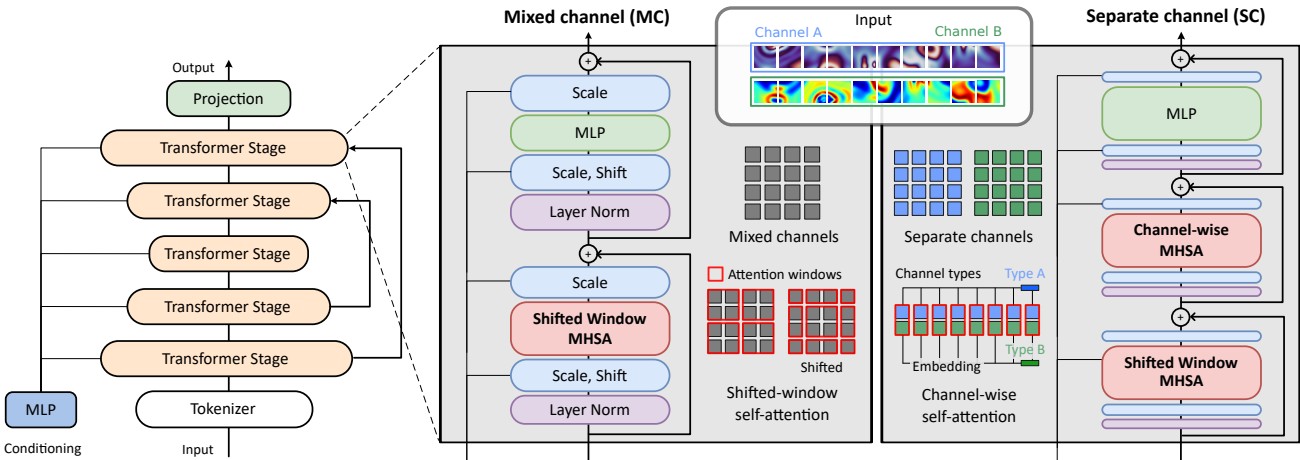

*Figure 2.* Architecture overview of PDE-Transformer. The multi-scale architecture combines up- and downsampling of tokens with skip connections between transformer stages of the same resolution. The attention operation is restricted to a local window of tokens. The window is shifted between two adjacent transformer blocks. Conditionings are embedded and used to scale and shift the intermediate token representations. The mixed channel (MC) version embeds different physical channels within the same token. The separate channel (SC) version embeds different physical channels independently. Tokens of different physical channels only interact via axial self-attention over the channel dimension. The types of channel (velocity, density, etc.) are part of the conditioning, which is distinct for each channel.

backbone based on an axial ViT, where axial attention is computed along the temporal and spatial dimensions.

**Learning for PDEs** While physical residuals pose difficult learning tasks (Raissi et al., 2019; Bruna et al., 2024), neural networks can also be combined with existing PDE solvers via learned corrections (Um et al., 2020; Dresdner et al., 2023; Thuerey et al., 2021), learned closures (Duraisamy et al., 2019; Sirignano & MacArt, 2023), or learned algorithmic components (Bar & Sochen, 2019; Kochkov et al., 2021), such as computation stencils (Bar-Sinai et al., 2019). Significant interest has been sparked in building neural operators (Lu et al., 2021; Kovachki et al., 2023), which map between function spaces. The attention mechanisms can be generalized to arbitrary meshes by including the distance between nodes as weighting, converging to an integral kernel operator as the resolution increases (Li et al., 2023a;b). As a result, transformers and attention can be broadly generalized, allowing them to work on arbitrary input and output points (Wu et al., 2024a). While enjoying beneficial theoretical properties, the added flexibility makes it difficult to scale to high resolutions and many points. Scalable operator transformer (Herde et al., 2024, scOT) is a hierarchical vision transformer with shifted windows for the attention computation. In contrast to scOT, PDE-Transformer is designed from the ground up as an improved diffusion transformer for PDEs, and significantly outperforms the former.

**Pre-training** Networks can be pre-trained using different strategies: autoregressive prediction (Radford et al., 2018),

masked reconstruction (Devlin et al., 2019; He et al., 2022) and contrastive learning (Chen et al., 2020). For PDEs, pre-training via autoregressive next-step prediction can include additional simulation parameters (Gupta & Brandstetter, 2023; Takamoto et al., 2023; Subramanian et al., 2023) or can be based on previous snapshots only (Herde et al., 2024). Pre-training has also been explored for PDEs in the context of neural operators (Li et al., 2021; Goswami et al., 2022; Wang et al., 2022), and domain transfers (Xu et al., 2023). Recently, models have also been pre-trained on multiple PDE dynamics at the same time (Subramanian et al., 2023; Yang et al., 2023a; McCabe et al., 2023).

## 3. PDE-Transformer

**Notation** We denote a spatiotemporal system by $S$. This system encompasses $n$ physical quantities $u(x, t) : \Omega_S \times [0, T] \to \mathbb{R}^n$ over a spatial domain $\Omega_S \subset \mathbb{R}^2$. We assume the data is discretized in time and space, i.e. the system is described by a sequence $[\mathbf{u}_0^S, \mathbf{u}_{\Delta t}^S, ..., \mathbf{u}_T^S]$, where each snapshot $\mathbf{u}_t^S$ is sampled on the points determined by the spatial discretization. We denote all additional information about S such as the type of PDE or simulation hyperparameters by the vector $\mathbf{c}$. The network, denoted $\mathcal{M}_\Theta$, is parameterized by weights $\theta$.

**Autoregressive Prediction** For autoregressive prediction tasks, the goal is to predict the snapshot $\mathbf{u}_t^S$ given a sequence of $T_\mathrm{p}$ preceding snapshots $\mathbf{u}_{t-T_\mathrm{p}\Delta t}^S, ..., \mathbf{u}_{t-\Delta t}^S$. Throughout the paper, we abbreviate $\mathbf{u}_\mathrm{in} = [\mathbf{u}_{t-T_\mathrm{p}\Delta t}^S, ..., \mathbf{u}_{t-\Delta t}^S]$ and

$\mathbf{u}_{\text{out}} = [\mathbf{u}_t^S]$ for autoregressive prediction. The definitions of $\mathbf{u}_{\text{in}}$ and $\mathbf{u}_{\text{out}}$ can be easily adapted to represent different tasks, e.g., $\mathbf{u}_{\text{in}} = \emptyset$ and $\mathbf{u}_{\text{out}} = [\mathbf{u}_t^S]$ for an unconditional generative model, or $\mathbf{u}_{\text{in}} = [\mathbf{u}_{t-\Delta t}^S, \mathbf{u}_{t+\Delta t}^S]$ and $\mathbf{u}_{\text{out}} = [\mathbf{u}_t^S]$ for temporal interpolation.

### 3.1. Design Space

We augment the diffusion transformer backbone (Peebles & Xie, 2023, DiT) to obtain PDE-Transformer. Its architecture is summarized in Figure 2. We describe the DiT architecture and our modifications step by step in the following.

**Patching**  DiT operates in a latent space with fixed dimension, whereas PDE-Transformer operates on the raw data. For data consisting of a single physical channel, given patch size $p$, an input of size $T \times H \times W$ is partitioned into $H/p \cdot W/p$ patches of size $T \times p \times p$. These patches are embedded into tokens representing a spatio-temporal subdomain via a linear layer that maps each patch to a $d$-dimensional vector. The patch size is a critical hyperparameter, as it directly controls the granularity of the token representation. By halving the patch size, the number of tokens quadruples, significantly increasing the required compute. We define an additional quantity, which we call the *expansion rate* $E(p) := d/(p^2 T)$ that specifies the rate at which the input data expands. Low expansion rates are better for scalability, as they coincide with fewer tokens. On the other hand, there is a clear correlation between patch size and the performance of the trained model, where lower patch sizes achieve better results. We explore this trade-off between accuracy and compute costs for physics simulations in Section 4.

**Multi-scale architecture**  While U-shaped architectures like the classic UNet (Ronneberger et al., 2015) have shown remarkable success, token down- and upsampling have not been used by DiTs in favor of a more streamlined architecture by the original authors (Peebles & Xie, 2023). The hierarchical structure of U-shaped architectures resembles the multi-scale nature of features in nature and adds a strong inductive bias. Several works (Bao et al., 2023; Hoogeboom et al., 2023; Tian et al., 2024) have combined U-shaped architectures with a transformer backbone architecture. We introduce *down-* and *upsample tokens* at the end of each transformer stage via PixelShuffle and PixelUnshuffle layers. In contrast to Bao et al. (2023) and Hoogeboom et al. (2023), we rely on adaptive layer normalization for conditioning. Tian et al. (2024) also feature a U-shaped design, but it is combined with token downsampling on the query-key-value tuple of the self-attention operation. We found that this slightly improves performance but comes at the cost of increased training and inference time, due to suboptimal accelerator utilization. Thus, we did not downsample tokens within the self-attention operation for PDE data in our implementation.

**Shifted Windows**  To prevent the quadratic blowup of the global self-attention in DiTs, we adopt the shifted window multi-head self-attention (MHSA) operation used by SwinTransformers (Liu et al., 2021), which limits the self-attention between tokens to a local window. A window with window size $w$ encompasses $w \times w$ spatio-temporal tokens at each block. To prevent discontinuities at the window borders between two adjacent layers, the windows are shifted by $w/2$ tokens. We evaluate the impact of the window size on the accuracy-compute tradeoff in Section 4. In contrast to DiTs, no absolute positions are added to the token embeddings. We use log-spaced relative positions of tokens within each window combined with a feed-forward neural network when computing attention scores (Liu et al., 2022). This improves translation-invariance for learning PDEs and increases generalization across different window resolutions.

**Mixed and separate channel representations**  PDEs can involve multiple physical quantities and hence require different numbers of physical *channels*, a fundamental difference to the fixed channel count of DiT architectures. The dynamics and scales of the physical channels can be fundamentally different. A naive strategy for this situation is to define a maximum number of channels $C_{\max}$ as an additional dimension of the input. Therefore, the embedding layer will map the spatio-temporal patch $T \times C_{\max} \times p \times p$ to the $d$-dimensional token embedding. If the data has fewer than $C_{\max}$ channels, they are padded with zeros. This is the **mixed channel (MC)** variant of PDE-Transformer. Effectively, this reduces the expansion rate $E(p)$ by the factor $1/C_{\max}$ and mixes channels with different physical meaning. This leads to an overly compressed token representation, which can deteriorate performance and lead to poor generalization capabilities. Instead, we propose to scale the compute with the number of channels and keep the expansion rate of each token the same *independent of the number of channels*. We achieve this by embedding each channel independently and by implementing interactions between tokens of different channels via an additional channel-wise axial MHSA operation in each block. The windowed self-attention is not computed between tokens of different channels. This variant is called **separate channel (SC)** below, and we evaluate both variants in detail for the pre-training dataset and when finetuning in Section 4.

**Conditioning mechanism**  Similar to DiTs, we use adaLN-Zero blocks (Goyal et al., 2017; Perez et al., 2018; Dhariwal & Nichol, 2021). The embedding vectors of all conditionings are added and vectors for scale and shift operations are regressed using a feed-forward network. In addition, the scale and shift vectors are initialized such that each residual block is equal to the identity function, which accelerates training. DiTs use the class label and diffusion

time as conditioning. When training PDE-Transformer as a diffusion model, we also include the diffusion time as conditioning. The class label corresponds to the PDE type in our setup. Moreover, for each physical channel in the SC version, we use a label embedding for the type of channel (e.g. density, vorticity). All labels use dropout with probability 10%, therefore our model can be used in both a conditional and unconditional setting. Extending this conditioning to additional simulation parameters is straightforward.

**Boundary conditions**   We explicitly consider periodic and non-periodic boundary conditions for the simulation domains. When shifting the attention windows, the tokens are rolled along the $x$- and $y$-axis as if they were aligned on a spatial grid matching their position. This mimics periodic boundary conditions for both axes. If periodicity is not required, it can be explicitly disabled in the architecture by masking token interactions in the computation of the attention scores.

**Algorithmic improvements**   We normalize Q and K of the self-attention operation using RMSNorm (Zhang & Sennrich, 2019) to avoid instabilities due to an uncontrolled growth of the attention entropy (Dehghani et al., 2023). In addition, we find that the training configuration of diffusion transformers from Peebles & Xie (2023) leads to training instabilities and suffers from spikes in the training loss for our datasets. Thus, we alter the learning rate from $1.0 \cdot 10^{-4}$ to $4.0 \cdot 10^{-5}$ and employ the AdamW optimizer using a small amount of weight decay with a factor of $10^{-15}$ for bf16-mixed precision training as recommended by Esser et al. (2024). Moreover, we find that gradient clipping based on the exponential moving average (EMA) of gradients prevents any remaining spikes in the loss curve and ensures stable training. Further details of the training approach can be found in Appendix A.

### 3.2. Supervised and Diffusion Training

**Supervised**   PDE-Transformer can be trained both in a supervised manner and as a diffusion model. For tasks with a deterministic solution, for example when training a surrogate for a deterministic solver, supervised training of PDE-Transformer with the MSE can be used, which allows for a fast inference in one step. In this case, the network is directly trained with the MSE loss

$$\mathcal{L}_S = \mathbb{E}\left[||\mathcal{M}_\Theta(\mathbf{u}_{\text{in}}, \mathbf{c}) - \mathbf{u}_{\text{out}}||_2^2\right]. \quad (1)$$

**Diffusion training**   If the solution is not deterministic, then diffusion training is preferable, as it enables sampling from the full posterior distribution, instead of learning an averaged solution. We adopt the flow matching (Lipman et al., 2023; Liu et al., 2023) formulation of diffusion models (Ho et al., 2020) for training. Given the input $\mathbf{u}_{\text{in}}$ and conditioning $\mathbf{c}$, samples $\mathbf{x}_0$ from a noise distribution $p_0 = \mathcal{N}(0, I)$

are mapped to samples $\mathbf{x}_1$ from the posterior $p_1$ via the ordinary differential equation (ODE) $d\mathbf{x}_t = v(\mathbf{x_t}, t)\, dt$. The network $\mathcal{M}_\Theta$ learns the velocity $v$ by regressing a vector field that generates a probability path between $p_0$ and $p_1$. Samples along the probability path are generated via the forward process

$$\mathbf{x}_t = t\, \mathbf{u}_{\text{out}} + [1 - (1 - \sigma_{\min})t]\, \epsilon \quad (2)$$

for $t \in [0, 1]$ with $\epsilon \sim \mathcal{N}(0, I)$ and a hyperparameter $\sigma_{\min} = 10^{-4}$. We denote $\mathbf{c}^t = [\mathbf{c}, t]$ and $\mathbf{u}_{\text{in}}^t = [\mathbf{u}_{\text{in}}, \mathbf{x}_t]$. The velocity $v$ can be regressed by training with the loss

$$\mathcal{L}_{\text{FM}} = \mathbb{E}\left[||\mathcal{M}_\Theta(\mathbf{u}_{\text{in}}^t, \mathbf{c}^t) - \mathbf{u}_{\text{out}} + (1 - \sigma_{\min})\epsilon||_2^2\right] \quad (3)$$

Once trained, samples from the posterior can be generated conditioned on $\mathbf{u}_{\text{in}}$ and $\mathbf{c}$ by sampling $\mathbf{x}_0 \sim \mathcal{N}(0, I)$ and solving the ODE $d\mathbf{x}_t = \mathcal{M}(\mathbf{u}_{\text{in}}^t, \mathbf{c}^t)\, dt$ from $t = 0$ until $t = 1$. We use the explicit Euler method and experiment with choosing a good step size $\Delta t$ for PDEs in Section 4.

## 4. Experiments

We evaluate the performance of PDE-Transformer for autoregressive prediction with $T_p = 1$ preceding snapshots. Our experiments are divided into two parts: first we compare PDE-Transformer to other SOTA transformer architectures on a large pre-training set of different PDEs, focusing on accuracy, training time and required compute. We motivate our design choices for PDE-Transformer in an ablation study. Second, we finetune the pre-trained network on three different challenging downstream tasks involving new boundary conditions, different resolutions, physical channels and domain sizes, showcasing its generalization capabilities to out-of-distribution data. Our models use three different configurations S, B and L that have different token embedding dimensions $d$ (96, 192 and 384 respectively). We use the name PDE-S to refer to PDE-Transformer with configuration S. Unless otherwise noted $p$ and $w$ are set to their default values $p = 4$ and $w = 8$ and the mixed channel (MC) version is used.

### 4.1. Pre-training dataset

**Training**   We train our models on a large set of 16 linear and non-linear PDEs, including Kolmogorov flow, Burgers' equation, different variants of the Gray Scott equation and many more. The datasets are based on APEBench (Koehler et al., 2024), and described in detail in Appendix C. For each type of PDE, we consider 600 trajectories of 30 simulation steps each, which are randomly split into a fixed training, validation and test set. The data is generated with a spectral solver at resolution $2048 \times 2048$ and downsampled to $256 \times 256$ with PDEs having either 1 or 2 physical channels. We apply gradient accumulation when training larger models to keep the batch size unchanged. For evaluation, we use an EMA of the models weights with a decay of 0.999.

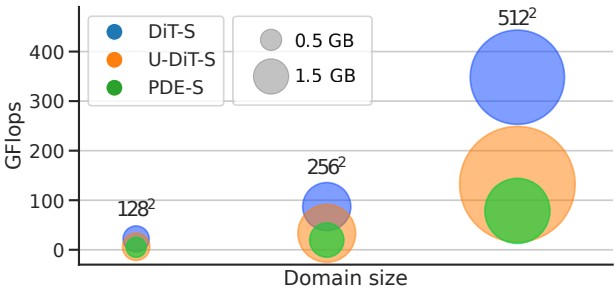

*Figure 3.* Scaling the domain size for patch size $p = 4$. An input resolution of $256 \times 256$ corresponds to $64 \times 64$ tokens. The area of a each point corresponds to the required GPU memory for inference (batch size 1).

**Evaluation metrics**  For the evaluation, we use the normalized RMSE, defined as

$$\text{nRMSE} = \frac{1}{M} \sum_{i=1}^{M} \sqrt{\frac{\text{MSE}(\hat{\mathbf{u}}_{\text{out}}, \mathbf{u}_{\text{out}})}{\text{MSE}(\mathbf{0}, \mathbf{u}_{\text{out}})}}, \qquad (4)$$

where $\hat{\mathbf{u}}_{\text{out}}$ is the network prediction and $M$ corresponds to the number of trajectories in the test dataset. We can autoregressively generate the entire trajectory for system $S$ and define the nRMSE at time $t$ comparing the predicted state $\hat{\mathbf{u}}_t^S$ at time $t$ with the reference $\mathbf{u}_t^S$. We average over all systems in the test datasets for each timestep.

**Windowed attention and multi-scale architecture**  We train several distinct and representative transformer models: DiT (Peebles & Xie, 2023) with no token down- or upsampling, UDiT (Tian et al., 2024) featuring a U-shape architecture, scOT (Herde et al., 2024), a neural operator transformer that includes a hierarchical architecture and shifted window attention, FactFormer (Li et al., 2023b) a transformer model based on axial factorized kernel integrals, a modern UNet (Ronneberger et al., 2015; Ho et al., 2020), and our proposed PDE-Transformer. When applicable, all models are trained with the S configuration of their architecture. We evaluate the trained models using the nRMSE and report values for the first step and after 10 steps of autoregressive prediction, see Table 1. We use a patch size of $p = 4$, resulting in $64 \times 64$ spatiotemporal tokens. UDiT-S and PDE-S achieve the best scores. Additional important differences between the two are visible in terms of required training time: first, PDE-S can be trained much faster (7h 42m). DiT-S has a much lower training time (13h 4m) compared to UDiT-S (18h 30m), even though the number of GFlops is higher. We believe this is caused by a better accelerator utilization of the DiT architecture on modern GPU hardware in comparison to UDiT. When training DiT-S, spikes in the loss curve occur that the model does not recover from. This happens repeatedly as the loss

decreases, irrespective of learning rate and gradient clipping strategies. This behavior indicates stability issues of the DiT architecture for the characteristics of PDE datasets. For evaluating DiT-S, we therefore use the checkpoint with the lowest validation loss.

*Table 1.* Training S configurations for 100 epochs on 4x H100 GPUs.

| Model | nRMSE$_1$ | nRMSE$_{10}$ | Time (h) | Params |
|---|---|---|---|---|
| DiT-S | 0.066 | 0.78 | 13h 4m | 39.8M |
| UDiT-S | **0.042** | 0.39 | 18h 30m | 58.9M |
| scOT-S | 0.051 | 0.59 | 21h 11m | 39.8M |
| FactFormer | 0.069 | 0.65 | 12h 25m | 3.8M |
| UNet[1] | 0.075 | 0.68 | 48h 00m | 35.7M |
| PDE-S | **0.044** | **0.36** | **7h 42m** | 33.2M |

**Efficient scaling of domain size and model parameters**  We compare the compute for inference (GFlops) and GPU memory requirements (GB) of PDE-S, UDiT-S and DiT-S for different input sizes assuming a patch size of $p = 4$. DiT-S is computationally very expensive for larger domains due to the global self-attention. UDiT-S scales better in general; however, GPU memory requirements scale badly due to convolutional layers within the attention operator for token up- and downsampling. PDE-S uses as few convolutional layers as possible for up- and downsampling within the U-shaped architecture, and achieves the best scaling for GFlops and GPU memory. We train the PDE-Transformer with configurations S, B and L. The overall architecture is kept identical, but we increase the dimension of the token embeddings $d$ with each configuration, see Table 2. Larger token embeddings lead to an improved performance. While the number of GFlops for inference increases by a factor of ca. 4x when doubling the token dimension, the total training time increases at a lower pace. This is due to an efficient accelerator utilization of matrix-multiplication in the self-attention operator on modern GPUs. The supervised and flow matching losses during training for the different configurations are shown in Figure 7 in the appendix.

**Axial attention for different physical channels**  We compare the mixed channel (MC) and separate channel (SC) versions of PDE-Transformer for all three configurations in Figure 4 (bottom). Most importantly, the flexibility of the proposed SC embeddings does not lead to any deterioration of inference accuracy. However, the compute of the SC variant increases, since the number of tokens scales linearly with the number of channels. Due to the fundamentally improved flexibility of the SC version we will focus on it in

---

[1]We train the UNet for 40 epochs, reaching a maximum compute budget of 2 days.

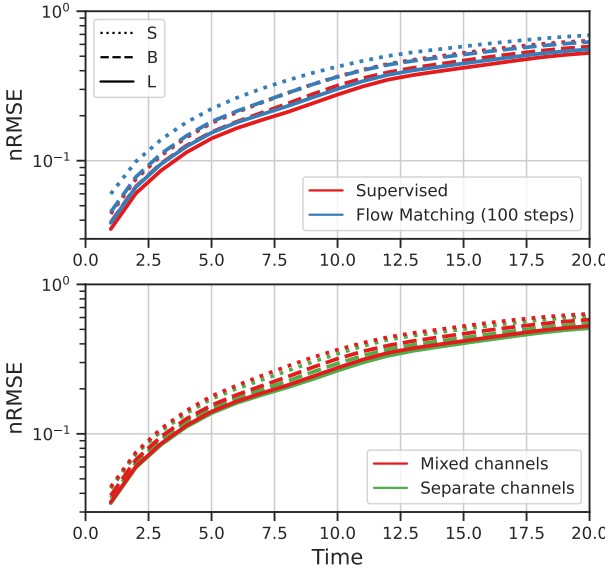

*Figure 4.* nRMSE evaluation of supervised training vs. training with flow matching and sampling from the posterior (top). Comparison of mixed channels (MC) vs. separate channel (SC) with axial attention over channel dimension (bottom).

the following, showing its improved versatility and generalization capabilities in the downstream tasks of Section 4.2.

**Supervision versus probabilistic learning**  We evaluate supervised training against training PDE-Transformer as a diffusion model via flow matching. While the diffusion PDE-Transformer samples from the posterior distribution, supervised training is incentivized to predict the mean of this posterior. Depending on the PDE, there are better suited evaluation metrics when considering a distribution of possible solutions, however our datasets comprises many different types of PDEs. Therefore, we only compare using the nRMSE, as shown at the top of Figure 4. As expected, the supervised training consistently outperforms samples from the diffusion version, since we consider only a single generated sample for each trajectory in the test set. Even though

*Table 2.* Scaling the token embedding dimension $d$. nRMSE$_1$ shows the results for the supervised training. The training time for 100 epochs is reported on 4x H100 GPUs.

| Config | $d$ | nRMSE$_1$ | Time (h) | GFlops |
|--------|-----|-----------|----------|--------|
| S | 96 | 0.045 | 7h 42m | 19.62 |
| B | 192 | 0.038 | 10h 40m | 76.55 |
| L | 384 | 0.035 | 20h 8m | 302.34 |

the diffusion version has a slightly lower accuracy in this evaluation, it is close to the supervised baseline. However, the ability to sample from the posterior, even at the cost of more network evaluations, is extremely useful in many downstream tasks and practical engineering applications. We also experiment with the step size $\Delta t$ for sampling, see Figure 8 in Appendix B and find that performance keeps improving with more steps.

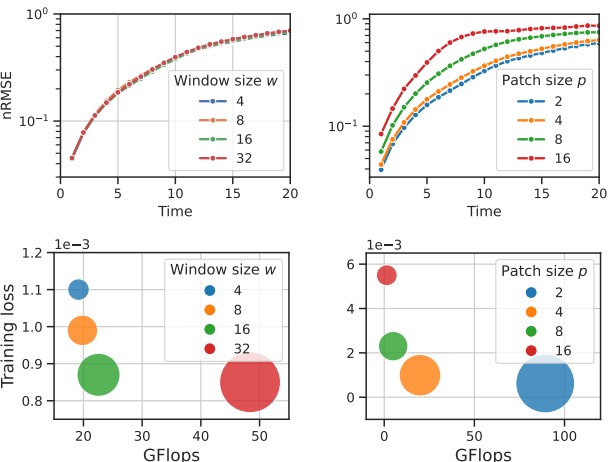

*Figure 5.* Impact of varying the window size $w$ (left) and patch size $p$ (right) on performance and required compute (GFlops). On the left, the patch size is kept at $p = 4$ and the self-attention window is increased. On the right, the patch size $p$ and window size $w$ are chosen such that the product of $p$ and $w$ is constant ($p \cdot w = 32$). Windows cover the same spatial domain.

**Influence of patch and window size**  We evaluate different patch and window sizes for PDE-Transformer. For this experiment, we focus on the S configuration with supervised training. We first test the effects of different window sizes $w$ while keeping the patch size $p$ fixed at $p = 4$. This means that the receptive field increases in the windowed self-attention, making interactions less local. While there is a decrease in training loss for larger window sizes, these lead to larger compute costs, as shown in Figure 5 on the left. At the same time, the nRMSE evaluation on the test set does not improve noticeably, indicating overfitting. We conclude that small window sizes are sufficient for the representative PDE behavior in our dataset. In the second experiment, we adjust window size and patch size at the same time. We keep the receptive field constant, i.e. when changing the patch size, we also modify the window size such that $p \cdot w = 32$ is constant, as shown in Figure 5 on the right. Smaller patch sizes lead to improved results, however they also significantly increase the required compute. There is a sweet spot for the accuracy-compute tradeoff at $p = 4$ and $w = 8$.

## 4.2. Downstream tasks

In this section, we evaluate the performance of our models on three challenging downstream tasks taken from the *Well* repository (Ohana et al., 2024): active matter, Rayleigh-Bénard convection (RBC), and shear flow. These tasks contain substantially more complex dynamics than any datasets seen during the pre-training phase, involving varied boundary conditions (periodic and non-periodic) and geometries (square and non-square domains). Detailed descriptions of these learning tasks are provided in Appendix E.1. In Section B.4 of the appendix, we also present a preliminary evaluation of the performance of our method in conjunction with low-rank adaption (Hu et al., 2022, LoRA).

For the downstream tasks, we compare PDE-Transformer to a range of SOTA models for PDE prediction: scOT-S (Herde et al., 2024), a Galerkin Transformer (Cao, 2021), OFormer (Li et al., 2023a), and a Fourier Neural Operator (Kossaifi et al., 2024, FNO). All baseline models are trained from scratch. For PDE-Transformer, we train two configuration S models: one initialized from scratch and the other from pre-trained weights. All models have a similar number of trainable parameters, except for OFormer, whose memory consumption increases significantly with model size. It is important to note that the scOT model and Galerkin Transformer are limited to square simulation domains, making them unsuitable for the RBC and shear flow tasks. Further details on each model can be found in Appendix A.3.

Figure 6 shows the results of this comparison. PDE-S with pre-training consistently yields more accurate predictions than the other baselines across the full range of difficult *Well* tasks. It provides an average improvement of 42% in terms of nRMSE, compared the second best model (FNO), demonstrating that PDE-Transformer consistently outperforms the range of baseline models. Notably, although pre-training was conducted with idealized data from a spectral solver, it improves the performance for all three downstream tasks featuring highly complex dynamics. In addition, while being pre-trained on periodic, square domains, the pre-training contributes to an improved performance on tasks with non-periodic boundary conditions (RBC) and non-square domains (RBC and shear flow). Detailed numerical values of the error during rollouts can be found in Appendix B.2. Additionally, note that the Poseidon model (Herde et al., 2024) provides pre-trained weights for the scOT model at configuration B. To enable a fair comparison, we evaluate the performance of our pre-trained PDE-B alongside the pre-trained scOT-B, as both models are of similar size. The PDE-B model demonstrates significant improvements in terms of accuracy, with an accumulated nRMSE reduction of 75% compared to Poseidon's scOT-B model, which becomes unstable for long-term rollouts. Figure 9 in the appendix presents the average rollout nRMSE of the model

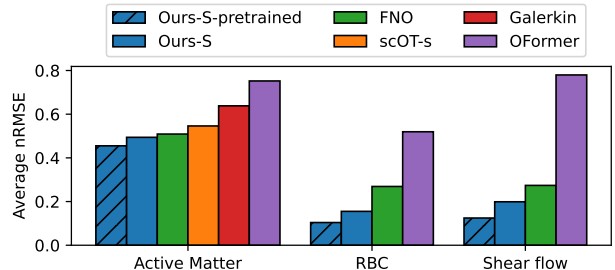

*Figure 6.* Average rollout nRMSE of different models on downstream tasks.

*Table 3.* Comparison of the PDE-Transformers with separate channel (SC) and mixed channel (MC) representations. The values are obtained by averaging the nRMSE of the first 20 rollout steps.

|        |    | Scratch | Pre-trained | Improvement |
|--------|----|---------|-------------|-------------|
| Active | SC | 0.494   | **0.455**   | **7.89%**   |
| Matter | MC | **0.493** | 0.479     | 2.84%       |
| RBC    | SC | 0.155   | **0.104**   | **32.90%**  |
|        | MC | **0.147** | 0.130     | 11.56%      |
| Shear  | SC | 0.199   | **0.125**   | **37.19%**  |
| Flow   | MC | **0.178** | 0.163     | 8.43%       |

predictions. Overall, these results highlight the benefits of the proposed architecture and pre-training approach.

In Table 3 we revisit the performance of the PDE-S using separate channel (SC) and mixed channel (MC) representations with the downstream tasks. While the performance of models without pre-training is similar, the SC version provides a greater performance boost on downstream tasks when pre-trained weights are used. Across the three challenging tasks, performance improvements with SC are $2.7\times$ to $4.4\times$ higher than the MC counterparts. This observation further highlights that the proposed channel-independent representation enables the network to retain more knowledge from pre-training without requiring extensive training or fine-tuning of the PDE-specific layers.

## 5. Limitations

PDE-Transformers is currently limited to 2D regular grids. Additionally, we focused on autoregressive prediction tasks. Testing and extending PDE-Transformer for tasks with noisy and only partially observed data, data assimilation, or inverse problems remains an open task for future work.

There are several directions for extending PDE-Transformer to unstructured grids. First, PDE-Transformer can be coupled with Graph Neural Operator (Li et al., 2020, GNO)

layers as the encoder and decoder, replacing the patchification, which map between the given geometry and a latent regular grid. Alternatively, it is also possible to generalize the notion of attention window to be defined on local neighbourhoods of graphs. Correspondingly, token up- and downsampling need to be replaced by respective graph pooling and upsampling operations.

## 6. Conclusion

We have introduced PDE-Transformer, a multi-purpose transformer model that addresses key challenges in physics-based simulations. Its multi-scale architecture modifies the diffusion transformer backbone and limits token interactions to a local window without sacrificing performance on learning PDE dynamics. PDE-Transformer outperforms state-of-the-art transformer architectures in terms of the accuracy-compute trade-off, demonstrating superior scaling for high-resolution data. By decoupling token embeddings for distinct physical channels, we further enhance performance, especially during fine-tuning on complex downstream tasks. Its focus on scalability makes PDE-Transformer an excellent backbone for foundation models for high-resolution simulation data. In the future, we aim to extend PDE-Transformer from 2D to 3D simulations.

## Acknowledgements

This work was supported by the ERC Consolidator Grant *SpaTe* (CoG-2019-863850). The authors gratefully acknowledge the scientific support and resources of the AI service infrastructure LRZ AI Systems provided by the Leibniz Supercomputing Centre (LRZ) of the Bavarian Academy of Sciences and Humanities (BAdW), funded by Bayerisches Staatsministerium für Wissenschaft und Kunst (StMWK).

## Impact Statement

This paper presents work whose goal is to advance the field of Machine Learning. There are many potential societal consequences of our work, none which we feel must be specifically highlighted here.

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

# A. Training configurations

## A.1. Training Hyperparameters

Table 4 summarizes the key hyperparameters used during training. We employ the Distributed Data Parallel (DDP) strategy, as supported by PyTorch Lightning (Falcon & The PyTorch Lightning team, 2019), to train models across multiple GPUs. To manage memory consumption effectively and maintain a consistent effective batch size across different hardware setups, we incorporate the gradient accumulation technique. Additionally, mixed-precision training is enabled for most models, except for the FNO and OFormer models in the downstream tasks. The exclusion is due to the inherent incompatibility of fast Fourier transformations within these architectures with the mixed-precision setup.

*Table 4.* Major hyperparameters for training.

|  | Pre-training | Downstream tasks |
|---|---|---|
| Effective batch size | 256 | 256 |
| Learning rate | $4.00 \cdot 10^{-5}$ | $1.00 \cdot 10^{-4}$ (Active matter & RBC) 
 $4.00 \cdot 10^{-5}$ (Shear flow) |
| Optimizer | AdamW | AdamW |
| Epochs | 100 | 2000 |

## A.2. EMA Gradient Clipping

In this study, we employ an exponential moving average (EMA) of the gradient norm to stabilize training. We compute EMA values for the gradient norm using two different coefficients, as outlined in Algorithm 1. The EMA with a larger coefficient places greater emphasis on historical values and serves as the clipping threshold. Meanwhile, the EMA with a smaller coefficient acts as the target value for gradient clipping. Notably, the clipping coefficient $\kappa$ must be set greater than 1 to ensure that $g_1$ effectively tracks changes in the gradient norm. Compared to traditional gradient clipping, which applies a fixed threshold and clipping value, EMA gradient clipping offers a more flexible approach by dynamically adjusting the threshold and clipping value. This adaptability is particularly advantageous, as gradient norms vary significantly across different model sizes and training stages.

---

**Algorithm 1** EMA Gradinet Clip.

**Input:** first EMA coefficient $\beta_1$, second EMA coefficient $\beta_2$, Clip threshold coefficient $\alpha$, Clip value coefficient $\kappa$.
Initialize $\beta_1 = 0.99$, $\beta_2 = 0.999$, $\alpha = 2$, $\kappa = 1.1$, $i = 0$, $g_1 = 0$, $g_2 = 0$.
**repeat**
  Get gradient $\mathbf{g}$ from training step
  **if** $i! = 0$ and $|\mathbf{g}| > \alpha \frac{g_2}{1-\beta_2^i}$ **then**
    $\mathbf{g} = \kappa \mathbf{g} \frac{g_1}{1-\beta_1^i}$
  **end if**
  $g_1 = \beta_1 g_1 + (1 - \beta_1)|\mathbf{g}|$
  $g_2 = \beta_1 g_2 + (1 - \beta_1)|\mathbf{g}|$
  $i = i + 1$
**until** training is finished

---

## A.3. Model Details

Table. 5 summarizes the sizes of the network in different models. The size definition of scOT models follows the definition in the Poseidon foundation model (Herde et al., 2024). Meanwhile, we rescale the size of FNO by setting the `n_modes_height=16`, `n_modes_width=16`, and `hidden_channels=192` in the official implementation (Kossaifi et al., 2024). For the Galerkin Transformer, we also change the `n_hidden=96`, `num_encoder_layers=5`, `dim_feedforward=128`, and `freq_dim=432` in the official implementation (Cao, 2021) to extend the network size. We noticed that the OFormer requires much more memory during the training compared with other models. We have to modify

*Table 5.* Size of the network of different models.

| Models | Number of trainable parameters |
|---|---|
| PDE-S | 46.57M |
| scOT-S | 39.90M |
| FNO | 42.72M |
| Galerkin Transformer | 38.82M |
| OFormer | 0.12M |
| PDE-B pre-trained | 178.97M |
| scOT-B pre-trained | 152.70M |

*Table 6.* Architecture hyperparameters of PDE-Transformer.

| Name of hyperparameter | Value |
|---|---|
| `window_size` | 8 |
| `depth` | [2, 4, 4, 6, 4, 4, 2] |
| `num_heads` | 16 |
| `mlp_ratio` | 4.0 |
| `class_dropout_prob` | 0.1 |
| `qkv_bias` | True |
| `activation` | GELU |

the `in_emb_dim=24` and `out_seq_emb_dim=48` of the original implementation (Li et al., 2023a) to make it have similar memory consumption as other models, although the network size is in different magnitude. We list hyperparameter for the architecture of PDE-Transformer in Table 6.

# B. Details and Additional Results

## B.1. Training Methodology

Details of the convergence of the training loss for different configurations of PDE Transformer can be found in Figure 7. The plots show that B and L configurations yield consistently lower training losses for both supervised and diffusion based training methodologies. Additionally, the left plot shows the stable behavior of the diffusion training with reduced loss fluctuations across epochs.

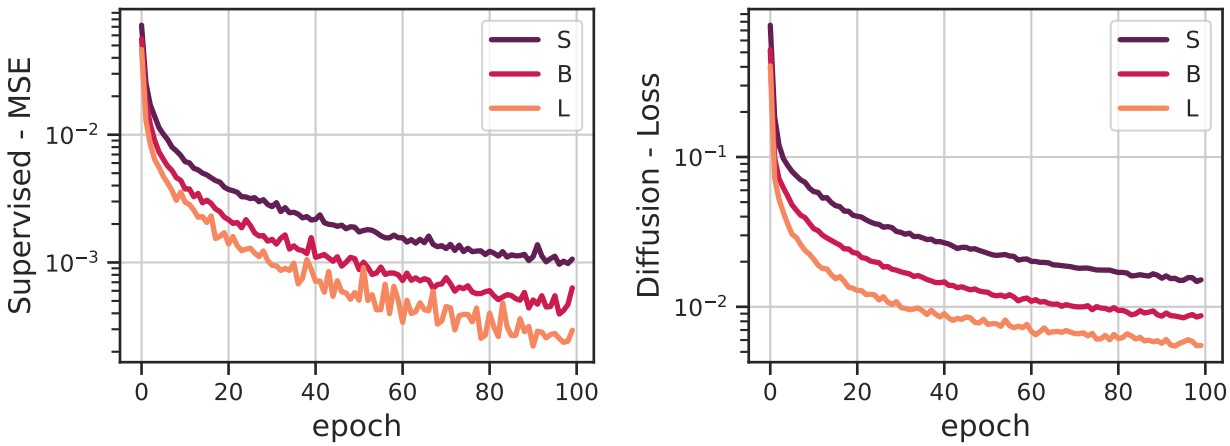

*Figure 7.* Training loss of supervised (left) and diffusion training (right) for configurations S, B and L.

Figure 8 shows the nRMSE over rollout steps for the PDE Transformer L configuration when using flow matching. After using more than ca. 20 flow integration steps the nRMSE values begin to stabilize at a lower level in terms of mean errors. Increasing the number of steps gives persistent improvements.

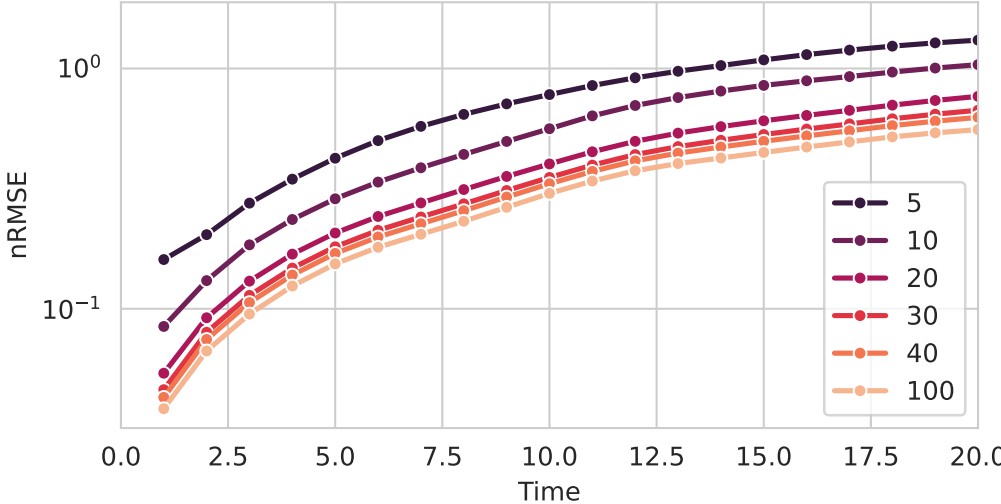

*Figure 8.* We evaluate the nRMSE against the number of inference steps for PDE-Transformer with configuration L. We use the explicit Euler method to solve the ODE for sampling.

## B.2. Downstream Tasks

Details of the comparison with the pre-trained Poseidon model scOT-B over time are shown in Figure 9. Both scOT-B and PDE-Transformer B have a similar size, show comparable performance during the very first frames of the trajectory. However, the prediction error of scOT-B accumulates rapidly over time, while our PDE-B produces stable trajectories with persistently low errors despite the challenging task. This is reflected in the snapshot for a case at $t = 8$ shown on the left of Figure 9. While scOT-B has diverged to a state dominated by oscillations, the PDE-B solution stays close to the reference solution.

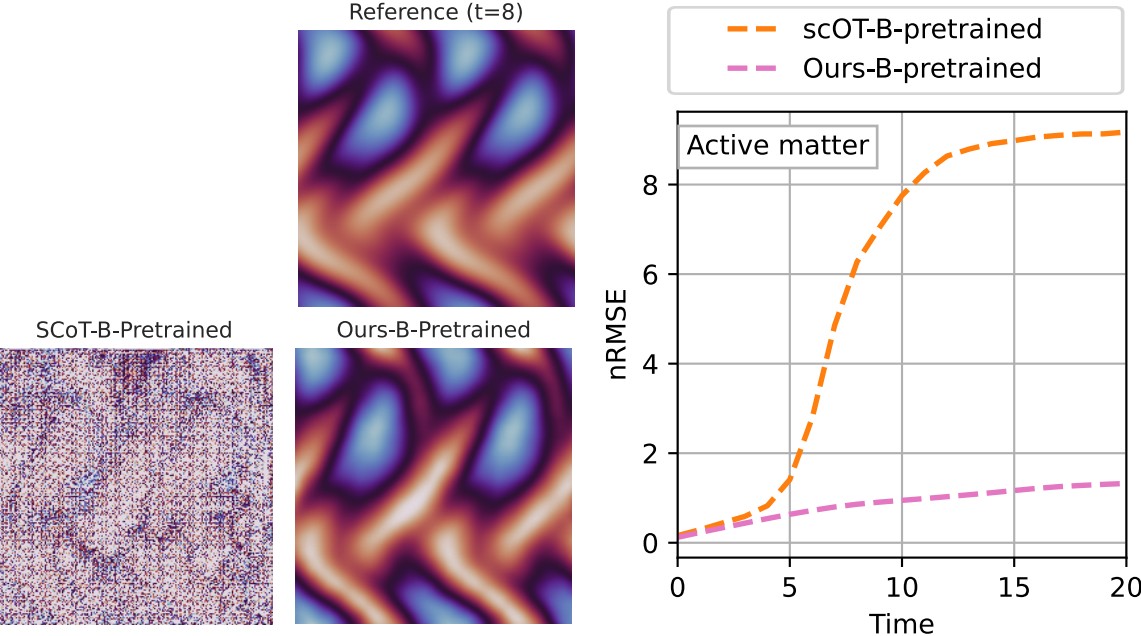

*Figure 9.* Average trajectory nRMSE of PDE-Transformer and scOT with pre-trained weights on active matter tasks. A representative frame at $f = 8$ is shown left.

## B.3. Numerical Values

Table 7 shows the exact numerical values of the nRMSE value in Figure 6. In addition, Figure 10 illustrates the nRMSE changes during the rollout for each downstream task. It highlights that the pre-trained PDE-Transformer consistently outperforms baselines over the course of the full rollout.

## B.4. Fine-tuning with LoRA

Low-Rank Adaptation (LoRA) (Hu et al., 2022) has recently emerged as a popular technique for addressing the computational challenges associated with fine-tuning foundation models (Zhang et al., 2024b; Wu et al., 2024b; Yang et al., 2024). Like many other parameter-efficient fine-tuning (PEFT) methodologies, LoRA significantly reduces the number of trainable parameters during fine-tuning, achieving this with minimal performance degradation. Building on its early success in fine-tuning large language models (Hu et al., 2022; Mao et al., 2025) and diffusion-based image models (Luo et al., 2023; Smith et al., 2024), LoRA has gained considerable traction in other domains. These include computer vision (Aleem et al., 2024; Lin et al., 2024), continual learning (Wistuba et al., 2023; Wei et al., 2025), and time-series prediction (Gupta et al., 2024b;a). Within the scientific deep-learning community, LoRA has also been applied to protein and materials engineering

*Table 7.* Average rollout nRMSE of different models on downstream tasks.

|  | Active Matter | RBC | Shear flow |
| --- | --- | --- | --- |
| FNO | 0.509 | 0.269 | 0.274 |
| OFormer | 0.752 | 0.520 | 0.780 |
| scOT-s | 0.546 | - | - |
| Galerkin | 0.638 | - | - |
| PDE-S | 0.494 | 0.155 | 0.199 |
| PDE-S pre-trained | **0.455** | **0.104** | **0.125** |

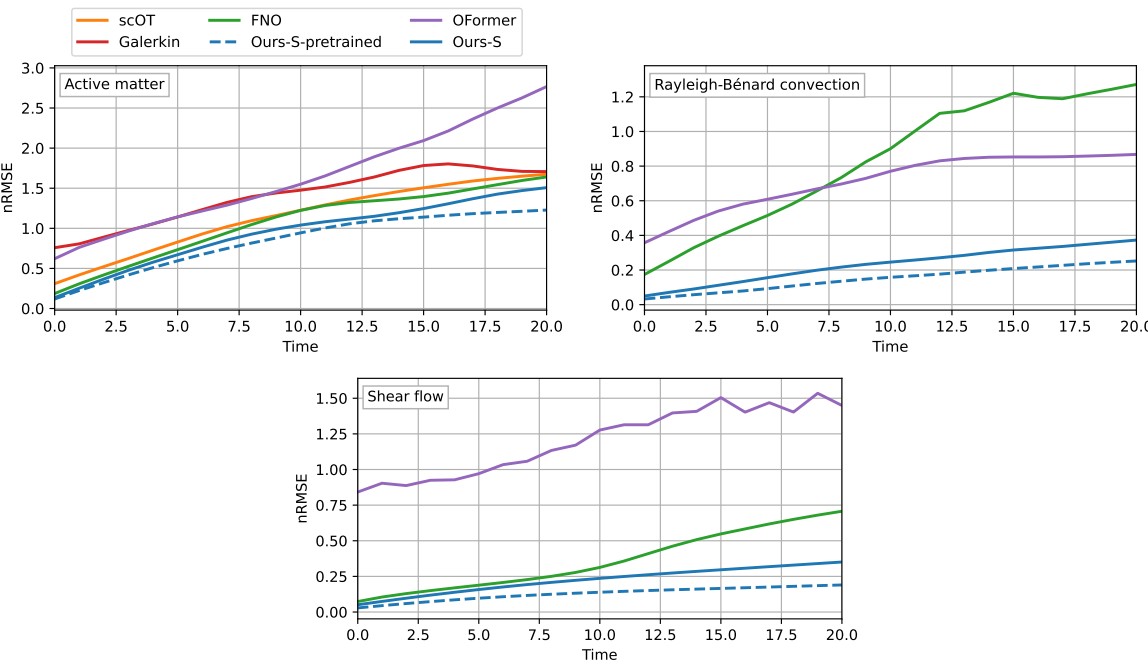

Figure 10. Average rollout nRMSE of different models on downstream tasks.

(Buehler & Buehler, 2024; Zeng et al., 2024). Despite these advancements, the application of LoRA in foundation models for partial differential equations (PDEs) remains relatively unexplored. Given the input of network $\mathbf{x}$ and the original pre-trined weight $\mathbf{W}_0 \in \mathbb{R}^{d \times k}$, LoRA introduces a low-rank matrix $\Delta \mathbf{W}$ to calculate the network output $\mathbf{y}$:

$$\mathbf{y} = \mathbf{W}_0 \mathbf{x} + \Delta \mathbf{W} \mathbf{x} = \mathbf{W}_0 \mathbf{x} + \frac{\alpha}{r} \mathbf{B} \mathbf{A} \mathbf{x}. \tag{5}$$

Here, $\mathbf{B} \in \mathbb{R}^{d \times r}$, $\mathbf{A} \in \mathbb{R}^{r \times k}$, $r$ is the rank, and $\alpha$ ($\alpha = r$ in the current study) is the coefficient used to rescale the low-rank matrix. By setting $r < dk/(d+k)$ and freezing the pre-trained weight during the fine-tuning, the trainable parameter during fine-tuning can be significantly reduced. We enable LoRA for the weights of linear layers and convolutional layers in the network and keep other parameters, e.g., biases, unchanged with their full parameters. Fig. 11 illustrates how the size of the network changes with different numbers of ranks. The size of PDE-Transformer-S, -B, and -L all grow linearly with more ranks. We evaluate the performance of LoRA on the downstream tasks by training a B-size PDE-Transformer with

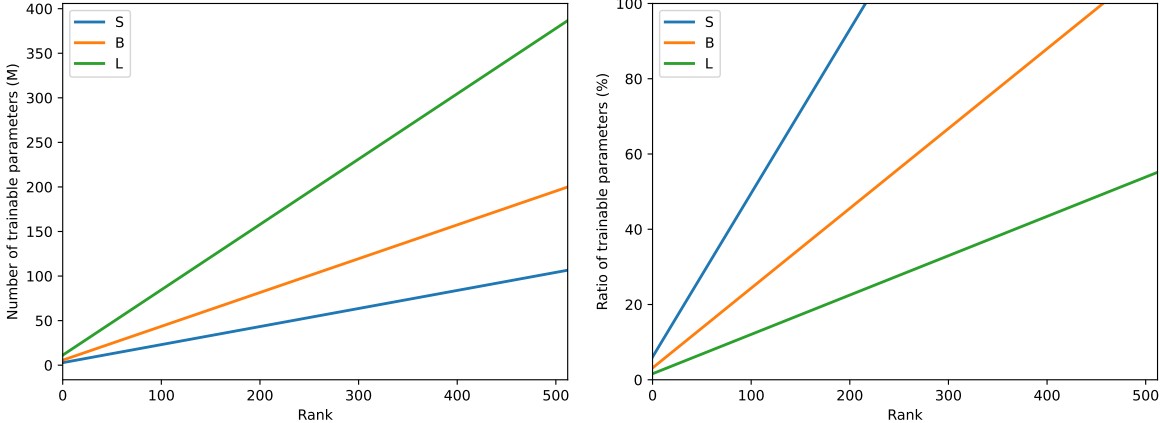

Figure 11. The number of trainable parameters of different sizes of PDE-Transformer w.r.t different LoRA ranks.

$r = 96$, which satisfies the theoretical minimum rank required for transformer-based models (Zeng & Lee, 2024). The final fine-tuned network consists of 42.06M trainable parameters, a size comparable to the S-size model. Figure 12 presents the average rollout performance of LoRA fine-tuning on the B-size model. In the active matter case, the LoRA-B model outperforms the LoRA-S model without pre-training, demonstrating advantages similar to those of the pre-trained S model. However, for the RBC task, the LoRA-B model performs comparably to the non-pre-trained model but falls short of the pre-trained model. These results indicate that while the LoRA fine-tuned model achieves performance similar to that of a model of the same size trained from scratch, it can not always reach the level of a fully pre-trained model with a similar size.

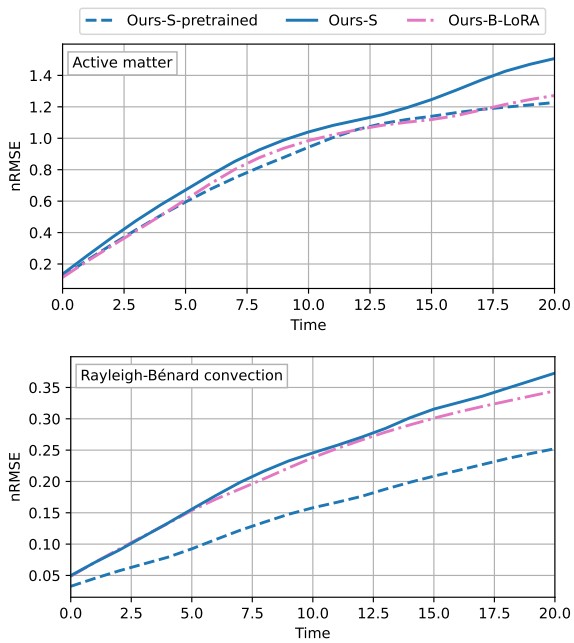

*Figure 12.* Performance of the LoRA finetuning on downstream tasks.

## C. Pre-training Datasets

The sources for our datasets are chosen to ensure a wide range of different PDEs, at high spatial resolutions, and with varying physical quantities across the simulations in each dataset. To simulate linear, reaction-diffusion, and nonlinear PDEs, we employ the *Exponax* solver (Koehler et al., 2024). It implements a range of Exponential Time Differencing Runge-Kutta (ETDRK) methods for numerically solving different PDEs in an efficient and unified manner. Our choice against using the authors benchmark APEBench directly was intentional, to ensure that the datasets are higher in resolution and more diverse in terms of physical behavior across simulations rather than only changing initial conditions. The ETDRK methods operate in Fourier space and as such do not allow for non-periodic domains or complex boundaries. We always use the physical solver interface across data sets, rather than the non-dimensionalized difficult-based interface, to provide a simpler, unified way of handling physical quantities for the task embedding.

### C.1. Linear PDEs

The *Exponax* solver framework features a wide range of Exponential Time Differencing Runge-Kutta (ETDRK) methods to efficiently simulate different PDEs (Koehler et al., 2024). The chosen linear PDEs are comparatively simple and have analytical solutions, but are nevertheless an important building block for more complex PDEs. Here, each linear PDE can be interpreted as a scalar quantity like density that is affected by different physical processes. Unless specified otherwise, sampling from intervals is always performed uniformly random below. Figure 13 shows example visualizations of every dataset described below.

In addition to varying physical parameters across the dataset, each simulation features randomized initial conditions. By default, the initial conditions are constructed in the following way: One of three initialization types with different spectral

*Table 8.* Overview of datasets simulated with *Exponax* (top, Koehler et al., 2024) featuring linear PDEs, reaction-diffusion PDEs, and nonlinear PDEs. The dimensionality of each dataset is described via the number of simulations $s$, time steps $t$, fields (or channels) $f$, and spatial dimensions $x$ and $y$. In addition to varying the specified quantities, the initial conditions of each simulation in $s$ are different.

| Dataset | $s$ | $t$ | $f$ | $x$ | $y$ | Varied Quantities across $s$ | Test Set |
|---|---|---|---|---|---|---|---|
| diff | 600 | 30 | 1 | 2048 | 2048 | viscosity $(x, y)$ | $s \in [500, 600[$ |
| fisher | 600 | 30 | 1 | 2048 | 2048 | diffusivity, reactivity | $s \in [500, 600[$ |
| sh | 600 | 30 | 1 | 2048 | 2048 | reactivity, critical number | $s \in [500, 600[$ |
| gs-alpha | 100 | 30 | 2 | 2048 | 2048 | initial conditions only | separate: $s = 30$, $t = 100$ |
| gs-beta | 100 | 30 | 2 | 2048 | 2048 | initial conditions only | separate: $s = 30$, $t = 100$ |
| gs-gamma | 100 | 30 | 2 | 2048 | 2048 | initial conditions only | separate: $s = 30$, $t = 100$ |
| gs-delta | 100 | 30 | 2 | 2048 | 2048 | initial conditions only | $s \in [80, 100[$ |
| gs-epsilon | 100 | 30 | 2 | 2048 | 2048 | initial conditions only | separate: $s = 30$, $t = 100$ |
| gs-theta | 100 | 30 | 2 | 2048 | 2048 | initial conditions only | $s \in [80, 100[$ |
| gs-iota | 100 | 30 | 2 | 2048 | 2048 | initial conditions only | $s \in [80, 100[$ |
| gs-kappa | 100 | 30 | 2 | 2048 | 2048 | initial conditions only | $s \in [80, 100[$ |
| burgers | 600 | 30 | 2 | 2048 | 2048 | viscosity | $s \in [500, 600[$ |
| kdv | 600 | 30 | 2 | 2048 | 2048 | domain extent, viscosity | $s \in [500, 600[$ |
| ks | 600 | 30 | 1 | 2048 | 2048 | domain extent | separate: $s = 50$, $t = 200$ |
| decay-turb | 600 | 30 | 1 | 2048 | 2048 | viscosity | separate: $s = 50$, $t = 200$ |
| kolm-flow | 600 | 30 | 1 | 2048 | 2048 | viscosity | separate: $s = 50$, $t = 200$ |

energy distributions implemented by Exponax is chosen uniformly random: First, the random truncated Fourier series initializer layers multiple Fourier series additively up to a cutoff at a certain frequency level. The cutoff threshold is chosen as a uniformly random integer from $[2, 11[$. Second, the Gaussian random field initializer creates a power-law spectrum in Fourier space, i.e., the energy decays polynomially with the wavenumber. The power-law exponent is chosen uniformly random from the interval $[2.3, 3.6[$. Third, the diffused noise initializer creates a tensor of values with white normally distributed noise, that is diffused afterwards. The resulting spectrum decays exponentially quadratic with an intensity rate chosen uniformly random from the interval $[0.00005, 0.01[$. For all initializers, the resulting values of the initial conditions are normalized to have a maximum absolute value of one after generation. For vector quantities, the randomly selected initializer is sampled independently for each vector component.

**Diffusion (`diff`)** features the spatial dissipation of density field due to diffusion. The diffusivity also known as viscosity depends on the coordinate axis direction in our setup. This process is visually similar to blurring the density field over time, as high frequency information is damped. While simple at first glance, diffusion processes occur across domains, for instance in physics, biology, economics, and statistics.

- Dimensionality: $s = 600$, $t = 30$, $f = 1$, $x = 2048$, $y = 2048$
- Initial Conditions: random truncated Fourier / Gaussian random field / diffused noise
- Boundary Conditions: periodic
- Time Step of Stored Data: 0.01
- Spatial Domain Size of Simulation: $[0, 1] \times [0, 1]$
- Fields: density
- Varied Parameters: viscosity $\in [0.005, 0.05[$ independently for $x, y$
- Validation Set: random 15% split of all sequences from $s \in [0, 500[$
- Test Set: all sequences from $s \in [500, 600[$

## C.2. Reaction-Diffusion PDEs

The *Exponax* solver (Koehler et al., 2024) was also used to simulate different reaction-diffusion PDEs. Such PDEs are most commonly encountered for local chemical reactions, but can also occur in domains such as biology, physics, or geology. They can be used to model traveling waves and pattern formation, and typically describe concentrations of one or more substances.

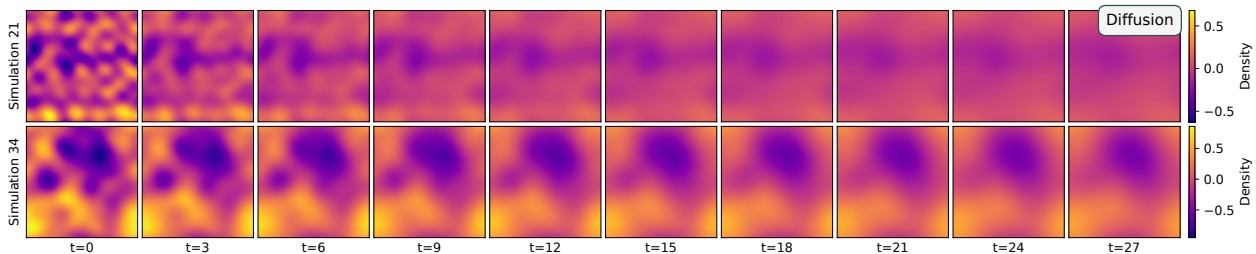

*Figure 13.* Random example simulations from `diff`.

**Fisher-KPP (`fisher`)** contains simulations of a reaction-diffusion system according to the Fisher-KPP equation. It describes how the concentration of a substance changes over time and space due to a reaction process controlled by a reactivity parameter, while also considering the spatial spread of the substance due to diffusion according to a diffusivity parameter. The equation can be applied in the context of wave propagation and population dynamics, as well as ecology or plasma physics. Figure 14 shows example visualizations from `fisher`.

- Dimensionality: $s = 600$, $t = 30$, $f = 1$, $x = 2048$, $y = 2048$
- Initial Conditions: random truncated Fourier / Gaussian random field / diffused noise (with clamping to $[0, 1]$)
- Boundary Conditions: periodic
- Time Step of Stored Data: 0.005
- Spatial Domain Size of Simulation: $[0, 1] \times [0, 1]$
- Fields: concentration
- Varied Parameters: diffusivity $\in [0.00005, 0.01[$ and reactivity $\in [5, 15[$
- Validation Set: random 15% split of all sequences from $s \in [0, 500[$
- Test Set: all sequences from $s \in [500, 600[$

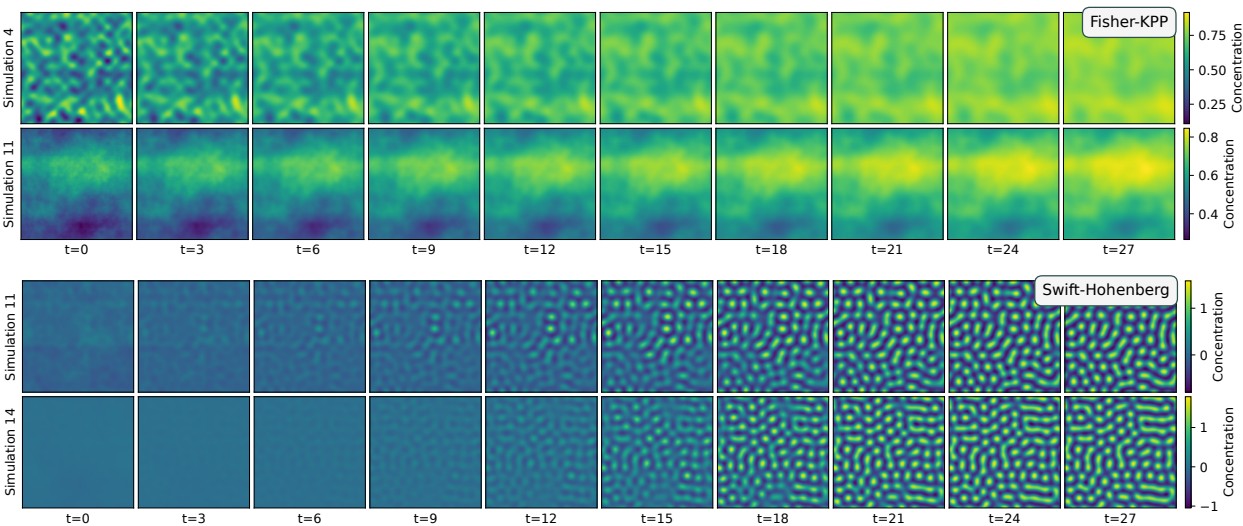

*Figure 14.* Random example simulations from `fisher` and `sh` (from top to bottom).

**Swift-Hohenberg (`sh`)** features simulations of the Swift-Hohenberg equation, which describes certain pattern formation processes. It can be applied to describe the morphology of wrinkles in curved elastic bilayer materials, for example, the formation of human fingerprints, where stresses between layers of skin lead to characteristic wrinkles. Figure 14 shows example visualizations from `sh`.

- Dimensionality: $s = 600$, $t = 30$, $f = 1$, $x = 2048$, $y = 2048$
- Initial Conditions: random truncated Fourier / Gaussian random field / diffused noise
- Boundary Conditions: periodic
- Time Step of Stored Data: 0.5 (with 5 substeps for the simulation)
- Spatial Domain Size of Simulation: $[0, 20\pi] \times [0, 20\pi]$
- Fields: concentration
- Varied Parameters: reactivity $\in [0.4, 1[$ and critical number $\in [0.8, 1.2[$
- Validation Set: random 15% split of all sequences from $s \in [0, 500[$
- Test Set: all sequences from $s \in [500, 600[$

**Gray-Scott (`gs`)**  describes a system in which two chemical substances react and diffuse over time. A substance $s_a$ with concentration $c_a$ is consumed by the reaction and is replenished according to a feed rate, while the product of the reaction $s_b$ with concentration $c_b$ is removed from the domain according to a kill rate. Depending on the configuration of both rates, simulations result in highly different steady or unsteady behavior with different patterns. Thus, we create various subsets: four with temporally steady configurations, which result in a state that does not substantially change anymore (`gs-delta`, `gs-theta`, `gs-iota`, and `gs-kappa`), and four temporally unsteady configurations, which continuously evolve over time (`gs-alpha`, `gs-beta`, `gs-gamma`, and `gs-epsilon`). For the unsteady case, separate test sets with longer temporal rollouts are created. Figure 15 shows example visualizations from the steady configurations, and Figures 16 and 17 from the unsteady configurations and corresponding test sets. For further details, we refer the reader to Pearson (1993). For all simulations, the diffusivity of the substances is fixed to $d_a = 0.00002$ and $d_b = 0.00001$. Furthermore, we use a random Gaussian blob initializer for these datasets: It creates four Gaussian blobs of random position and variance in the center 60% (20% for `gs-kappa`) of the domain, where the initialization of $c_a$ is the complement of $c_b$, i.e. $c_a = 1 - c_b$.

**Steady Configurations (`gs-delta`, `gs-theta`, `gs-iota`, and `gs-kappa`):**

- Dimensionality: $s = 100$, $t = 30$, $f = 2$, $x = 2048$, $y = 2048$ (per configuration)
- Initial Conditions: random Gaussian blobs
- Boundary Conditions: periodic
- Time Step of Simulation: 1.0 (all configurations)
- Time Step of Stored Data:
    - `gs-delta`: 130.0
    - `gs-theta`: 200.0
    - `gs-iota`: 240.0
    - `gs-kappa`: 300.0
- Number of Warmup Steps (discarded, in time step of data storage):
    - `gs-delta`: 0
    - `gs-theta`: 0
    - `gs-iota`: 0
    - `gs-kappa`: 15
- Spatial Domain Size of Simulation: $[0, 2.5] \times [0, 2.5]$
- Fields: concentration $c_a$, concentration $c_b$
- Varied Parameters: feed rate and kill rate determined by configuration (i.e., initial conditions only within configuration)
    - `gs-delta`: feed rate: 0.028, kill rate: 0.056
    - `gs-theta`: feed rate: 0.040, kill rate: 0.060
    - `gs-iota`: feed rate: 0.050, kill rate: 0.0605
    - `gs-kappa`: feed rate: 0.052, kill rate: 0.063

- Validation Set: random $15\%$ split of all sequences from $s \in [0, 80[$
- Test Set: all sequences from $s \in [80, 100[$

**Unsteady Configurations (`gs-alpha`, `gs-beta`, `gs-gamma`, and `gs-epsilon`):**

- Dimensionality: $s = 100$, $t = 30$, $f = 2$, $x = 2048$, $y = 2048$ (per configuration)
- Initial Conditions: random Gaussian blobs
- Boundary Conditions: periodic
- Time Step of Simulation: 1.0 (all configurations)
- Time Step of Stored Data:
    - `gs-alpha`: 30.0
    - `gs-beta`: 30.0
    - `gs-gamma`: 75.0
    - `gs-epsilon`: 15.0
- Number of Warmup Steps (discarded, in time step of data storage):
    - `gs-alpha`: 75
    - `gs-beta`: 50
    - `gs-gamma`: 70
    - `gs-epsilon`: 300
- Spatial Domain Size of Simulation: $[0, 2.5] \times [0, 2.5]$
- Fields: concentration $c_a$, concentration $c_b$
- Varied Parameters: feed rate and kill rate determined by configuration (i.e., initial conditions only within configuration)
    - `gs-alpha`: feed rate: 0.008, kill rate: 0.046
    - `gs-beta`: feed rate: 0.020, kill rate: 0.046
    - `gs-gamma`: feed rate: 0.024, kill rate: 0.056
    - `gs-epsilon`: feed rate: 0.020, kill rate: 0.056
- Validation Set: random $15\%$ split of all sequences from $s \in [0, 100[$
- Test Set: separate simulations with $s = 30$, $t = 100$, $f = 2$, $x = 2048$, $y = 2048$ (per configuration)

### C.3. Nonlinear PDEs

Nonlinear PDEs are generally difficult to study, as even the question of the existence of analytical solutions is already a hard problem. Furthermore, most general techniques do not work across cases, and single nonlinear PDEs are commonly tackled as individual problems. The *Exponax* solver (Koehler et al., 2024) provides tools to approach some nonlinear PDEs, however, we also consider other data sources below.

**Burgers (`burgers`)** features simulations of Burgers' equation, which is similar to an advection-diffusion problem. Rather than the transport of a scalar density, it describes how a flow field itself changes due advection and diffusion. This can lead to the development of sharp discontinuities or shock waves, making it difficult to simulate accurately. Burgers' equation also has applications in nonlinear acoustics and traffic flow. Figure 18 shows example visualizations from `burgers`.

- Dimensionality: $s = 600$, $t = 30$, $f = 2$, $x = 2048$, $y = 2048$
- Initial Conditions: random truncated Fourier / Gaussian random field / diffused noise
- Boundary Conditions: periodic
- Time Step of Stored Data: 0.01 (with 50 substeps for the simulation)
- Spatial Domain Size of Simulation: $[0, 1] \times [0, 1]$

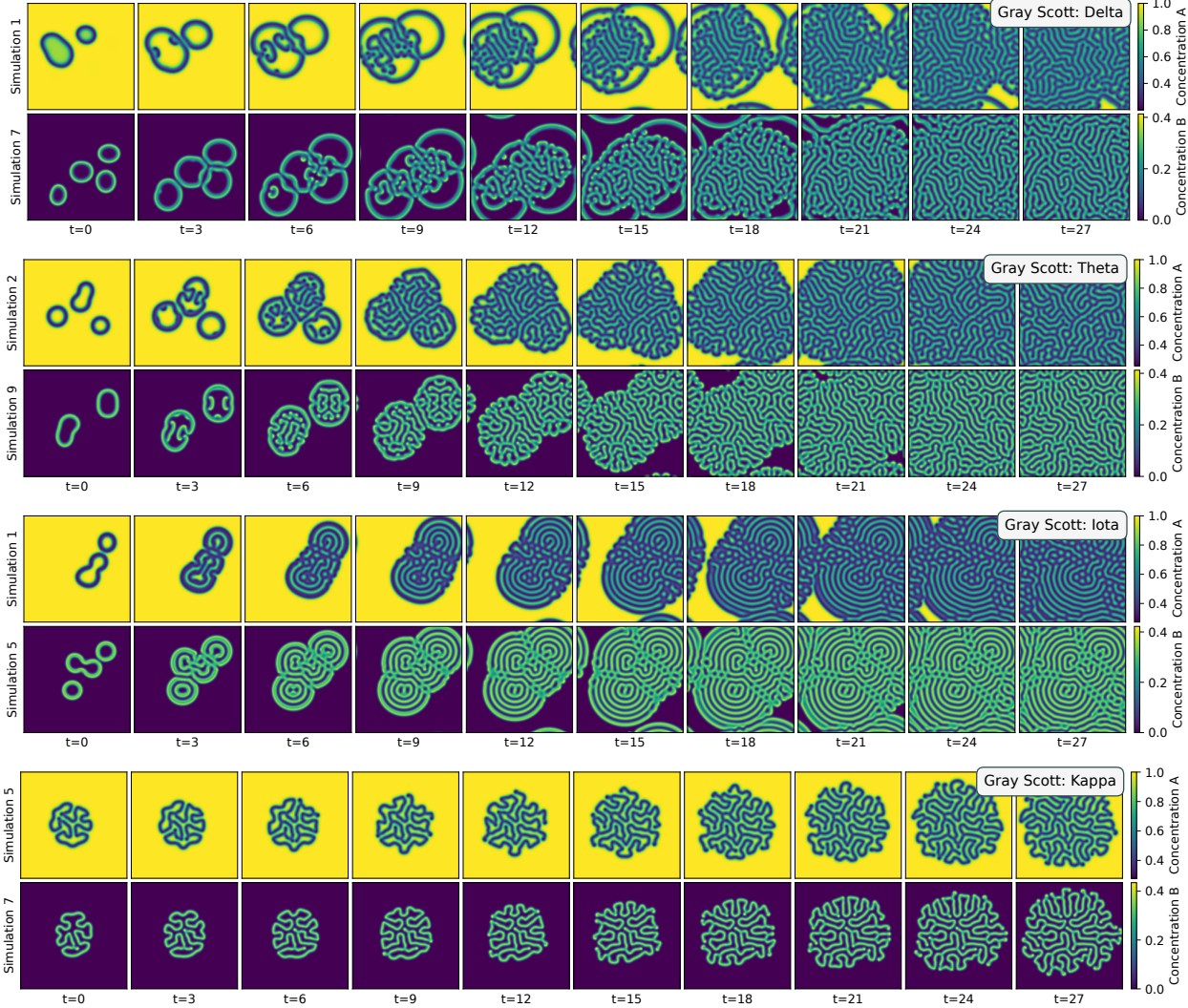

*Figure 15.* Random example simulations from steady configurations of the Gray-Scott model of a reaction-diffusion system: `gs-delta`, `gs-theta`, `gs-iota`, and `gs-kappa`.

- Fields: velocity $(x, y)$
- Varied Parameters: viscosity $\in [0.00005, 0.0003[$
- Validation Set: random $15\%$ split of all sequences from $s \in [0, 500[$
- Test Set: all sequences from $s \in [500, 600[$

**Korteweg-de-Vries (`kdv`)**   contains simulations of the Korteweg-de-Vries equation on a periodic domain, which serves as a model of waves on shallow water. It is challenging as energy is transported to high spatial frequencies, leading to individual moving solition waves with unchanged shape and propagation speed. Across simulations, the convection coefficient with a value of -6, and the dispersivity coefficient with a value of 1 remain constant. Figure 18 shows example visualizations from `kdv`.

- Dimensionality: $s = 600$, $t = 30$, $f = 2$, $x = 2048$, $y = 2048$
- Initial Conditions: random truncated Fourier / Gaussian random field / diffused noise
- Boundary Conditions: periodic

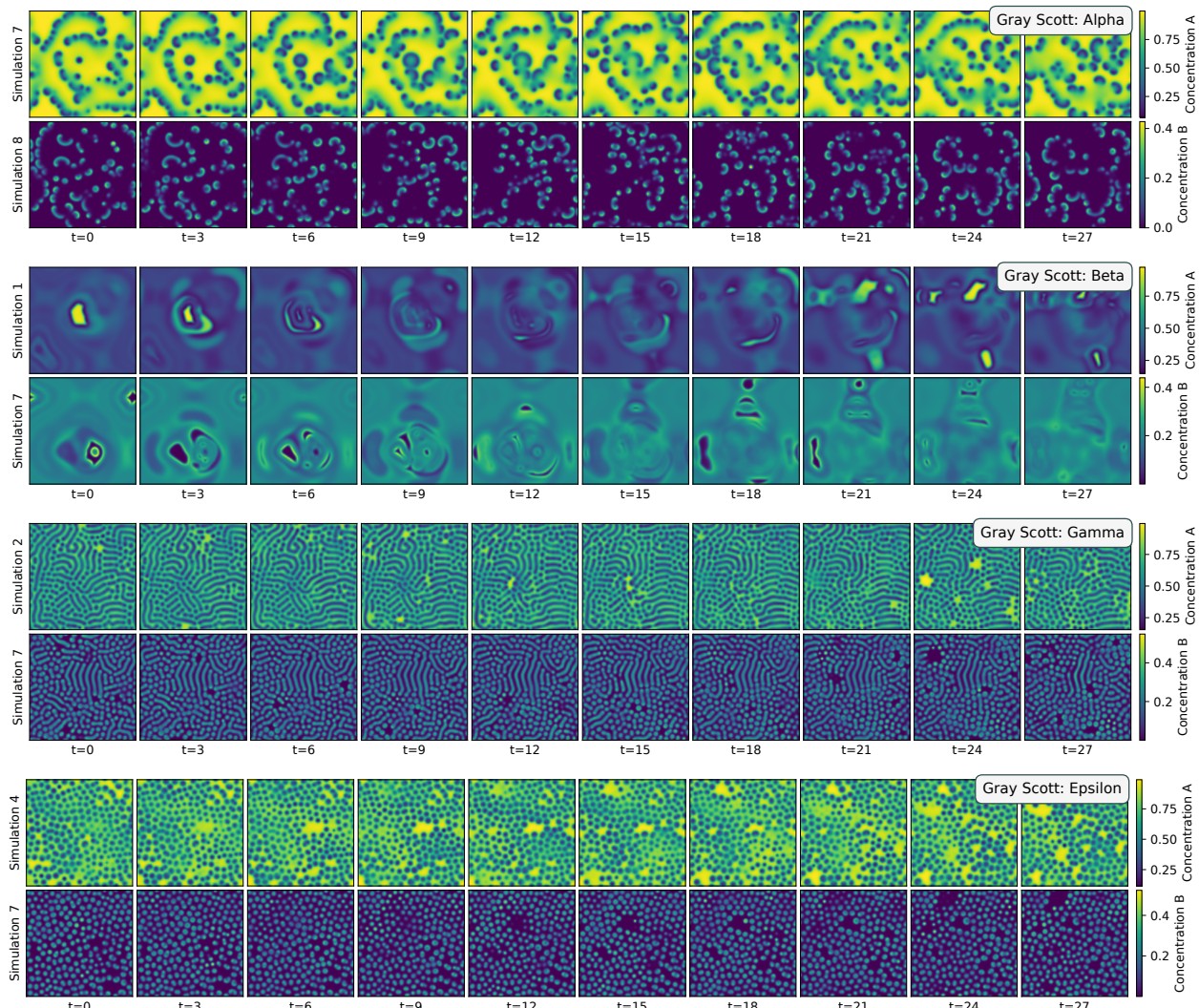

*Figure 16.* Random example simulations from unsteady configurations of the Gray-Scott model of a reaction-diffusion system: `gs-alpha`, `gs-beta`, `gs-gamma`, and `gs-epsilon`.

- Time Step of Stored Data: 0.05 (with 10 substeps for the simulation)
- Spatial Domain Size of Simulation: varied per simulation
- Fields: velocity $(x, y)$
- Varied Parameters: domain extent $\in [30, 120[$ identically for $x, y$, i.e. a square domain, and viscosity $\in [0.00005, 0.001[$
- Validation Set: random $15\%$ split of all sequences from $s \in [0, 500[$
- Test Set: all sequences from $s \in [500, 600[$

**Kuramoto-Sivashinsky (`ks`)**   features simulations of the Kuramoto-Sivashinsky equations on a periodic domain, which models thermo-diffusive flame instabilities in combustion. It also has applications in reaction-diffusion systems. The equation is well-known for its chaotic behavior, where temporal trajectories with slightly different initial conditions can substantially diverge over time. The initial transient phase of the simulations is discarded. For the `ks` dataset, a test set with longer rollout is used to investigate how well models can deal with this chaotic behavior. Figure 18 shows example visualizations from `ks`.

- Dimensionality: $s = 600, t = 30, f = 1, x = 2048, y = 2048$

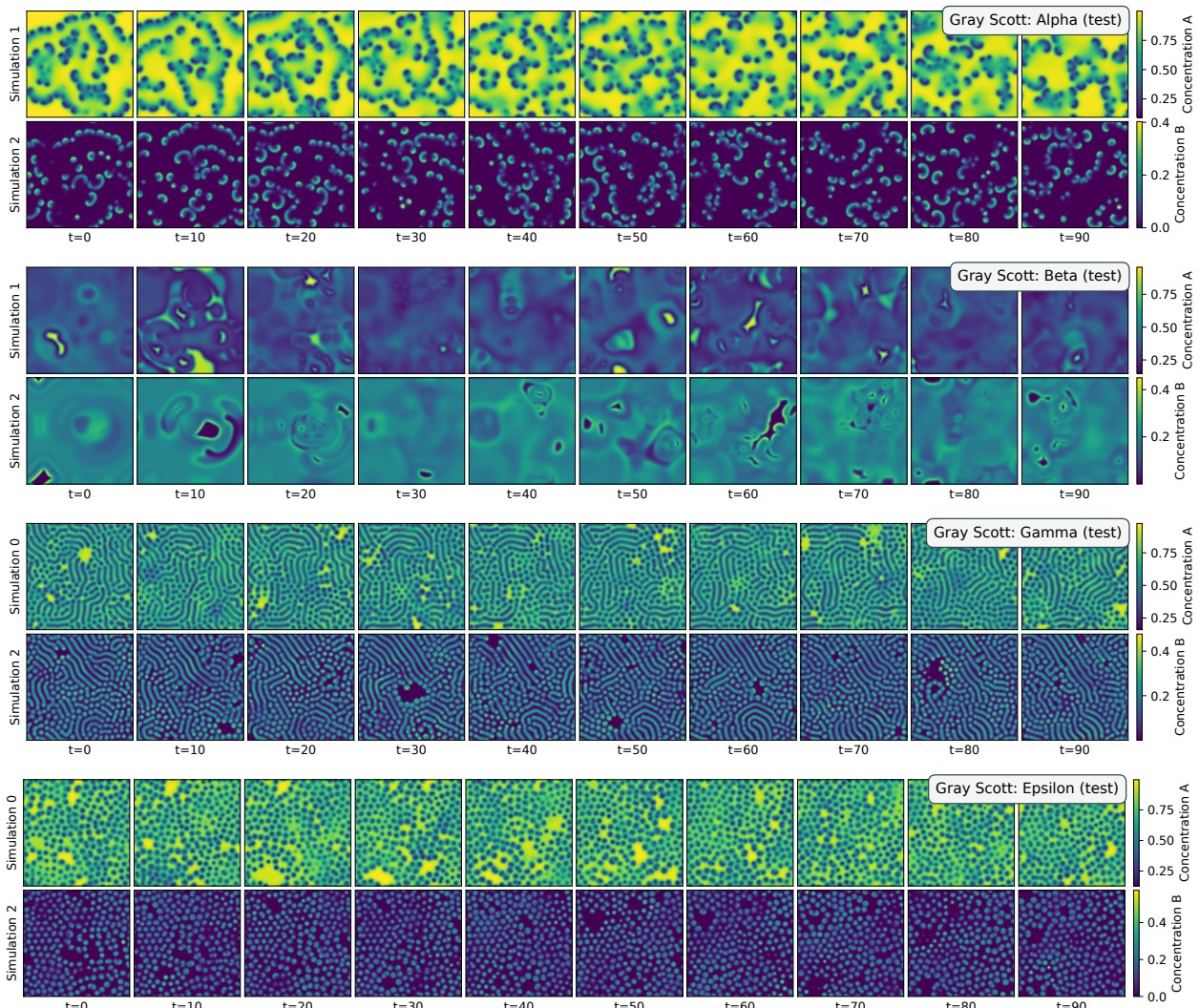

*Figure 17.* Random example simulations from test sets with longer rollout from `gs-alpha`, `gs-beta`, `gs-gamma`, and `gs-epsilon`.

- Initial Conditions: random truncated Fourier / Gaussian random field / diffused noise
- Boundary Conditions: periodic
- Time Step of Stored Data: 0.5 (with 5 substeps for the simulation)
- Number of Warmup Steps (discarded, in time step of data storage): 200
- Spatial Domain Size of Simulation: varied per simulation
- Fields: density
- Varied Parameters: domain extent $\in [10, 130[$ identically for $x, y$, i.e. a square domain
- Validation Set: random 15% split of all sequences from $s \in [0, 600[$
- Test Set: separate simulations with $s = 50$, $t = 200$, $f = 1$, $x = 2048$, $y = 2048$

**Decaying Turbulence (`decay-turb`)**    contains simulations of the Navier-Stokes equations in a streamfunction-vorticity formulation on a periodic domain. The simulations exhibit swirling turbulent vortices that decay over time. For this dataset, a test set with longer rollout is used, where the decay over time is even more pronounced. Figure 19 shows example visualizations from `decay-turb`.

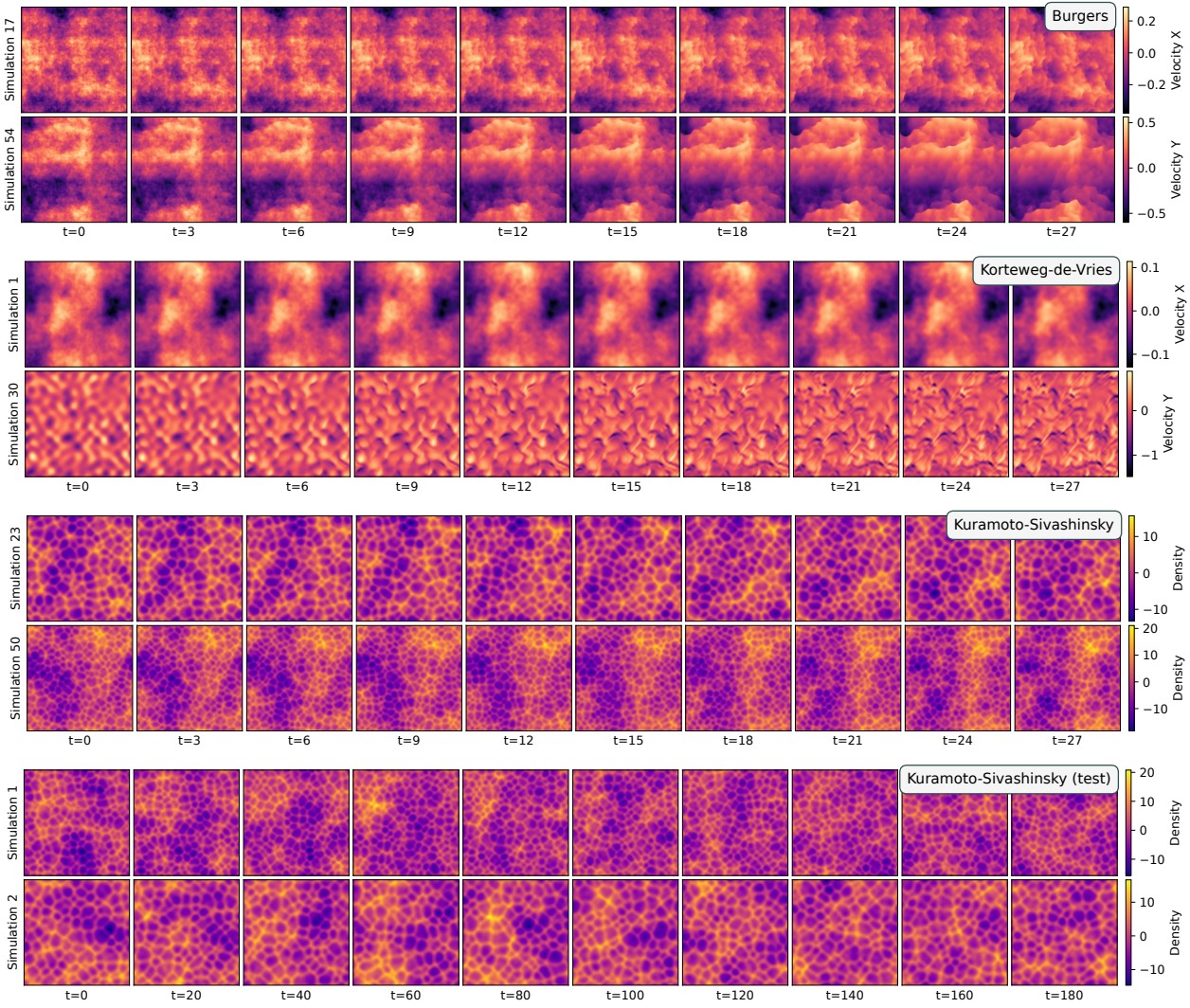

*Figure 18.* Random example simulations from `burgers`, `kdv`, `ks`, and the test set of `ks` with longer rollout.

- Dimensionality: $s = 600$, $t = 30$, $f = 1$, $x = 2048$, $y = 2048$
- Initial Conditions: random truncated Fourier / Gaussian random field / diffused noise
- Boundary Conditions: periodic
- Time Step of Stored Data: 3.0 (with 500 substeps for the simulation)
- Spatial Domain Size of Simulation: $[0, 1] \times [0, 1]$
- Fields: vorticity
- Varied Parameters: viscosity $\in [0.00005, 0.0001[$
- Validation Set: random $15\%$ split of all sequences from $s \in [0, 600[$
- Test Set: separate simulations with $s = 50$, $t = 200$, $f = 1$, $x = 2048$, $y = 2048$

**Kolmogorov Flow (`kolm-flow`)** features simulations of the Navier-Stokes equations in a streamfunction-vorticity formulation on a periodic domain. In contrast to the decaying turbulence above, an additional forcing term ensures that new energy is introduced into the system that sustains the vortices indefinitely, leading to spatiotemporal chaotic behavior. The transient phase of the simulations where the initialization transforms to a stripe pattern and into vortices afterwards is

discarded. For this dataset, a test set with longer rollout is used to test how well models can deal with the chaotic behavior of the flow. Figure 19 shows example visualizations from `kolm-flow`.

- Dimensionality: $s = 600$, $t = 30$, $f = 1$, $x = 2048$, $y = 2048$
- Initial Conditions: random truncated Fourier / Gaussian random field / diffused noise
- Boundary Conditions: periodic
- Time Step of Stored Data: 0.3 (with 1500 substeps for the simulation)
- Number of Warmup Steps (discarded, in time step of data storage): 50
- Spatial Domain Size of Simulation: $[0, 1] \times [0, 1]$
- Fields: vorticity
- Varied Parameters: viscosity $\in [0.0001, 0.001[$
- Validation Set: random $15\%$ split of all sequences from $s \in [0, 600[$
- Test Set: separate simulations with $s = 50$, $t = 200$, $f = 1$, $x = 2048$, $y = 2048$

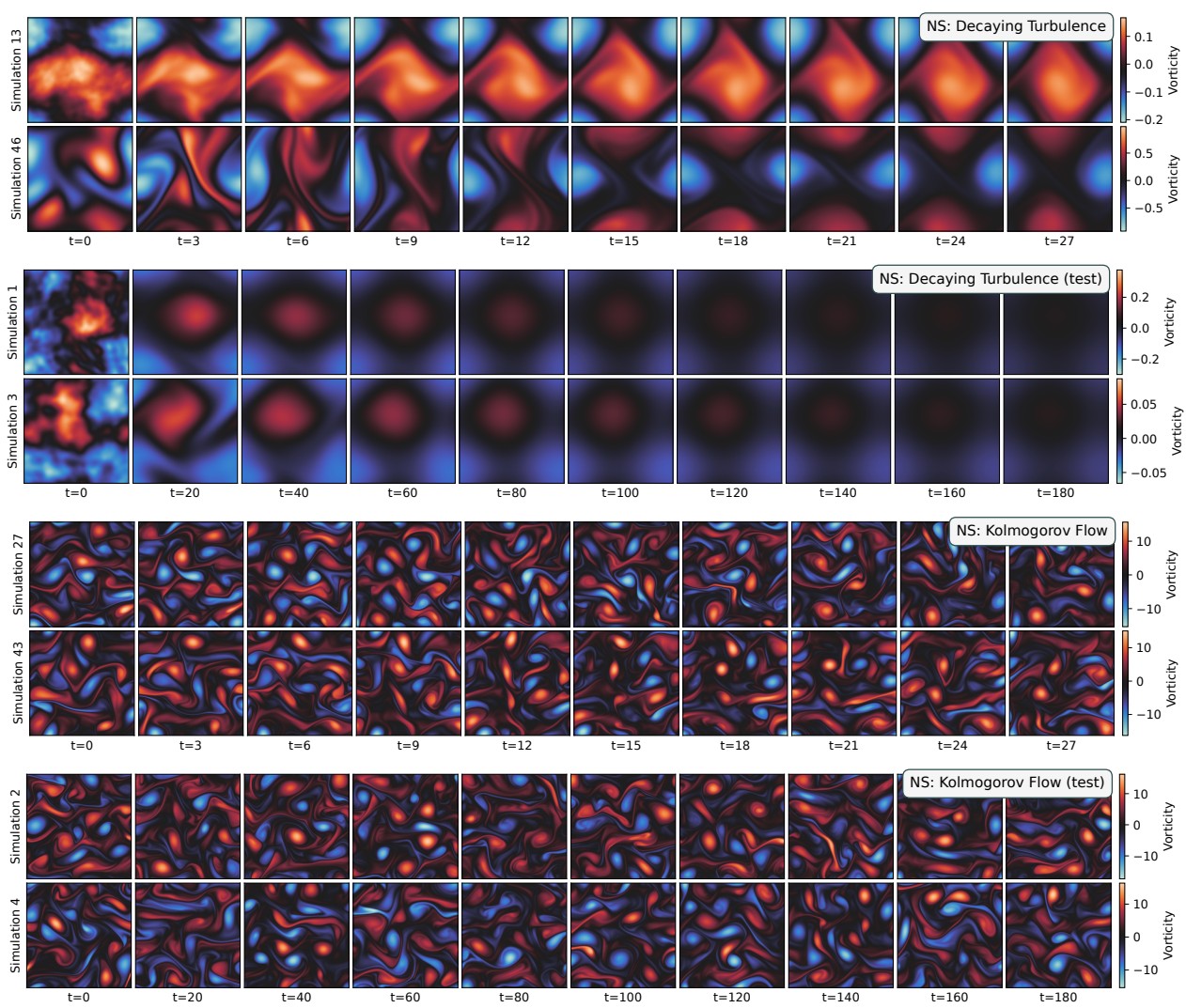

*Figure 19.* Random example simulations from `decay-turb`, and `kolm-flow`, with examples from each corresponding test set with longer rollout.

## D. Autoregressive Predictions

We show results for all datasets in Table 9 in Table 10 for the nRMSE with rollouts of 1 step and 20 steps respectively. Autoregressive predictions of trajectories on the test datasets from $t = 0$ until $t = 27$ are visualized below, see Figures 20 to 23, for PDE-L trained with the MSE loss and mixed channels (MC).

*Table 9.* nRMSE$_1$ after 1 step for the pre-training datasets.

| Method | PDE Dataset | | | | | | | | | | | | | | | |
|---|---|---|---|---|---|---|---|---|---|---|---|---|---|---|---|---|
| | diff | burgers | kdv | ks | fisher | gs-alpha | gs-beta | gs-gamma | gs-delta | gs-epsilon | gs-theta | gs-iota | gs-kappa | sh | decay-turb | kolm-flow |
| DiT-S | 0.0528 | 0.0262 | 0.0553 | 0.0609 | 0.0310 | 0.0388 | 0.0405 | 0.0942 | 0.0475 | 0.0284 | 0.0402 | 0.0365 | 0.0556 | 0.0856 | 0.2570 | 0.1209 |
| UDiT-S | 0.0370 | 0.0191 | **0.0435** | 0.0178 | 0.0242 | 0.0224 | 0.0300 | 0.0302 | 0.0149 | 0.0161 | 0.0125 | 0.0150 | 0.0193 | **0.0519** | 0.2487 | 0.0715 |
| scOT-S | 0.0674 | 0.0358 | 0.0536 | 0.0240 | **0.0230** | 0.0254 | 0.0400 | 0.0449 | 0.0271 | 0.0232 | 0.0212 | 0.0215 | 0.0311 | 0.0589 | **0.2114** | 0.0987 |
| FactFormer | 0.1440 | 0.0455 | 0.0823 | 0.0407 | 0.0231 | 0.0256 | 0.0347 | 0.0547 | 0.0343 | 0.0172 | 0.0255 | 0.0244 | 0.0410 | 0.0816 | 0.2248 | 0.1441 |
| UNet | 0.0559 | 0.0392 | 0.0606 | 0.0469 | 0.0335 | 0.0441 | 0.0575 | 0.0845 | 0.0413 | 0.0416 | 0.0308 | 0.0355 | 0.0429 | 0.0829 | 0.2885 | 0.2385 |
| PDE-S | 0.0370 | 0.0215 | 0.0480 | 0.0216 | 0.0247 | 0.0248 | 0.0295 | 0.0316 | 0.0172 | 0.0175 | 0.0129 | 0.0141 | 0.0231 | 0.0651 | 0.2361 | 0.0873 |
| PDE-B | **0.0348** | 0.0162 | 0.0456 | 0.0145 | **0.0230** | 0.0270 | 0.0298 | 0.0256 | 0.0141 | 0.0140 | 0.0092 | 0.0095 | 0.0165 | 0.0640 | 0.2206 | 0.0578 |
| PDE-L | 0.0349 | **0.0135** | 0.0455 | **0.0111** | 0.0235 | **0.0196** | **0.0260** | **0.0184** | **0.0113** | **0.0076** | **0.0063** | **0.0064** | **0.0134** | 0.0647 | **0.2113** | **0.0447** |

*Table 10.* nRMSE$_{20}$ after 20 steps for the pre-training datasets.

| Method | PDE Dataset | | | | | | | | | | | | | | | |
|---|---|---|---|---|---|---|---|---|---|---|---|---|---|---|---|---|
| | diff | burgers | kdv | ks | fisher | gs-alpha | gs-beta | gs-gamma | gs-delta | gs-epsilon | gs-theta | gs-iota | gs-kappa | sh | decay-turb | kolm-flow |
| DiT-S | 0.2677 | 0.8003 | 0.4915 | 1.6441 | 0.7041 | 1.4206 | 1.1469 | 1.0784 | 1.5140 | 1.2283 | 1.5285 | 1.2335 | 1.2387 | 0.9253 | 0.8148 | 1.2348 |
| UDiT-S | **0.2035** | 0.1982 | 0.3157 | 0.9865 | 0.6144 | 0.6983 | 0.7592 | 0.7958 | 1.0605 | 0.3591 | 1.0323 | 0.9341 | 0.7869 | 0.6882 | 1.0018 | 0.8575 |
| scOT-S | 0.8773 | 0.3948 | 0.4351 | 1.1377 | 0.7017 | 0.8794 | 0.9355 | 0.9100 | 1.1182 | 0.4704 | 1.0945 | 1.0103 | 1.0584 | 0.6872 | 1.0455 | 0.9866 |
| FactFormer | 1.9913 | 0.6704 | 0.8224 | 1.3988 | 0.6370 | 0.8579 | 0.8107 | 0.9878 | 1.1288 | 0.5425 | 1.0834 | 1.0160 | 1.2319 | 0.8426 | 1.2972 | 1.1670 |
| UNet | 0.5263 | 0.7142 | 0.5829 | 1.3061 | 0.7327 | 1.0608 | 1.1585 | 0.9748 | 1.0159 | 0.7716 | 0.9879 | 0.9197 | 0.9803 | 0.8322 | 0.9746 | 1.2597 |
| PDE-S | 0.2095 | 0.1879 | 0.3383 | 0.9741 | **0.5689** | 0.6696 | 0.6643 | 0.7887 | 1.0686 | 0.3196 | 1.0247 | 0.7601 | 0.7272 | 0.6757 | 0.9141 | 0.9370 |
| PDE-B | 0.2090 | 0.1142 | 0.3204 | 0.8599 | 0.6131 | 0.6186 | 0.6110 | 0.7558 | 1.0253 | 0.3636 | 0.9902 | 0.6432 | 0.6435 | 0.6354 | 0.7333 | 0.8005 |
| PDE-L | 0.2104 | **0.0921** | **0.3004** | **0.7357** | 0.5731 | **0.5554** | **0.5924** | **0.7083** | **0.9916** | **0.1997** | **0.9382** | **0.5333** | **0.5572** | **0.6112** | **0.7011** | **0.7102** |

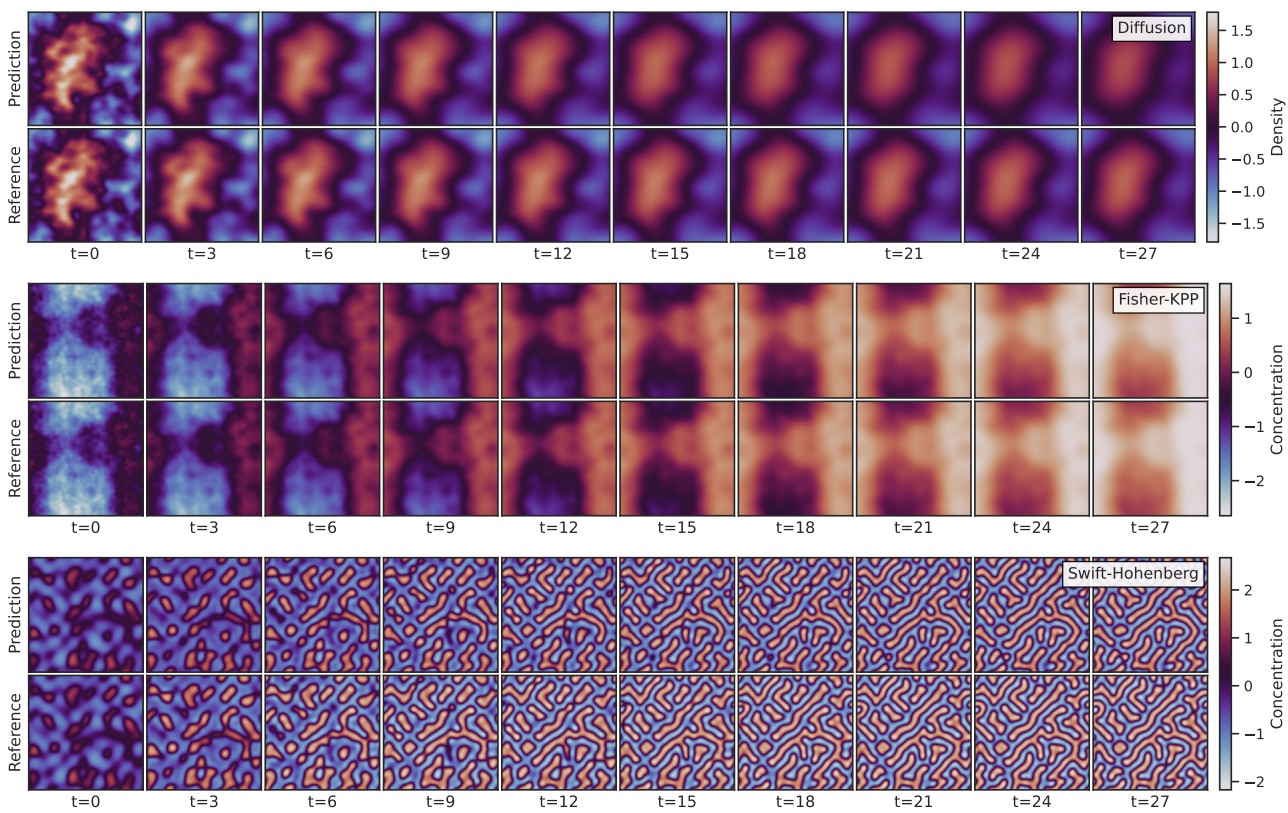

*Figure 20.* Visualizations of model's prediction on `diff`, `fisher` and `sh`.

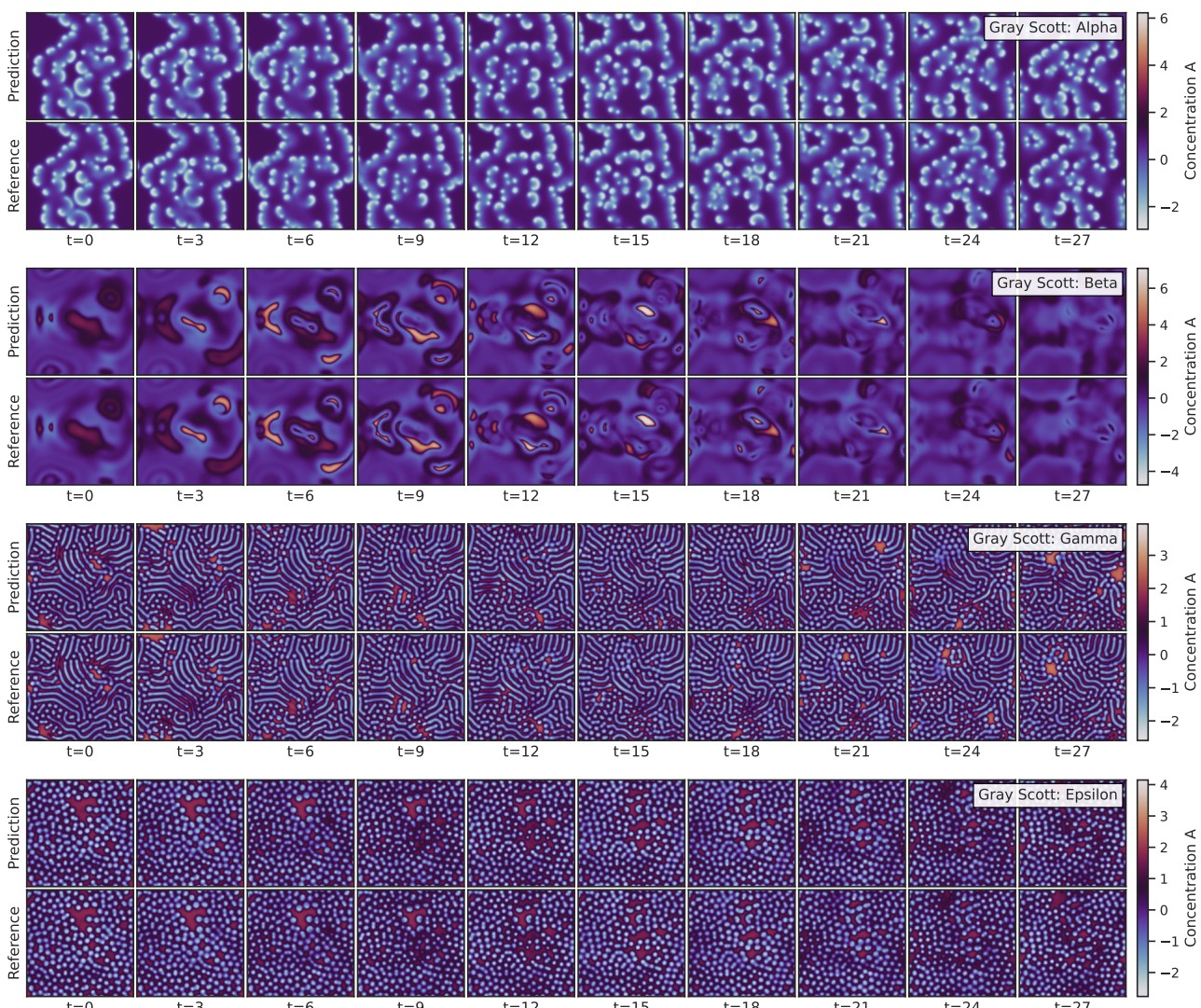

*Figure 21.* Visualizations of model's prediction on `gs-alpha`, `gs-beta`, `gs-gamma` and `gs-epsilon`.

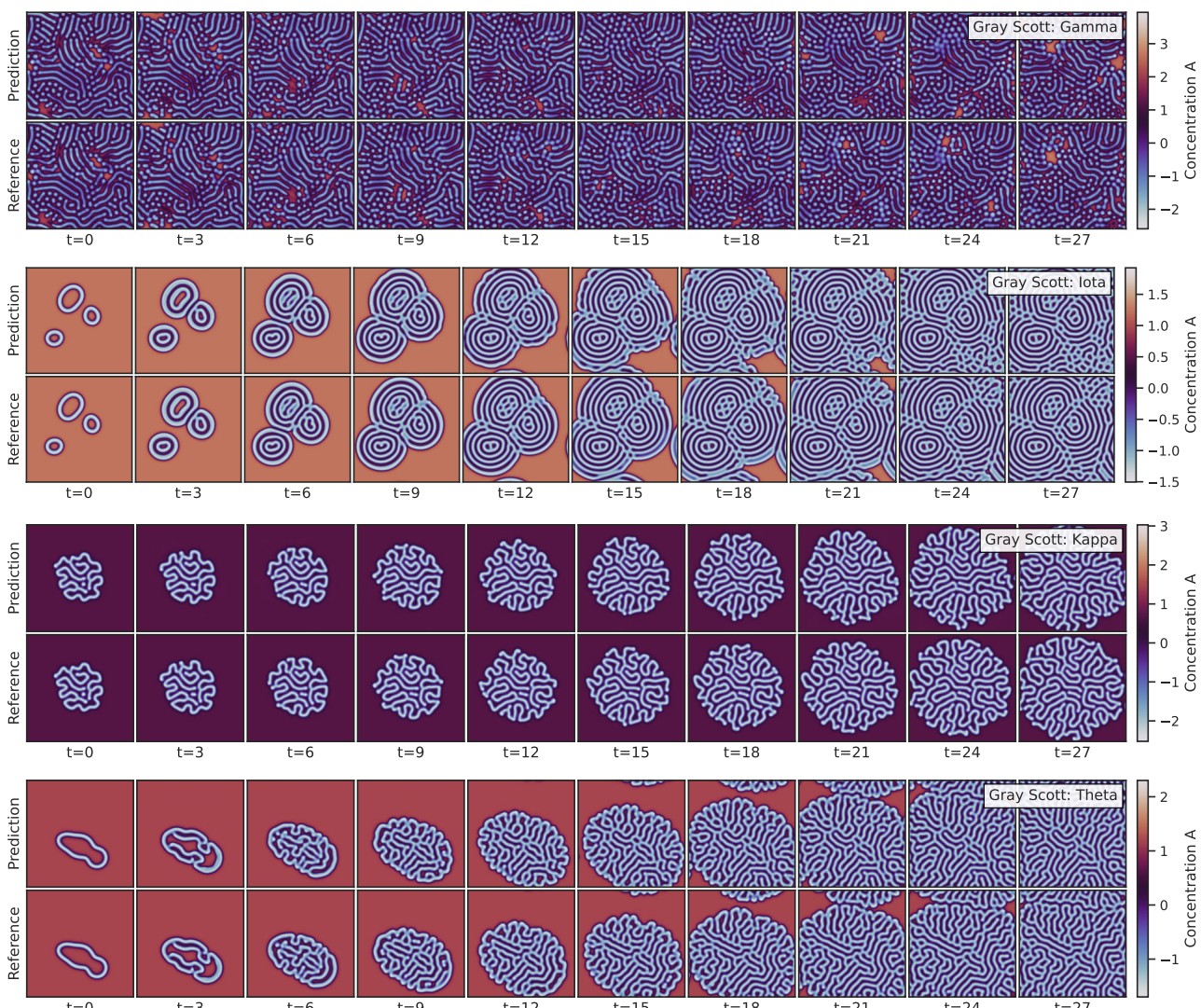

*Figure 22.* Visualizations of model's prediction on `gs-gamma`, `gs-iota`, `gs-kappa` and `gs-theta`.

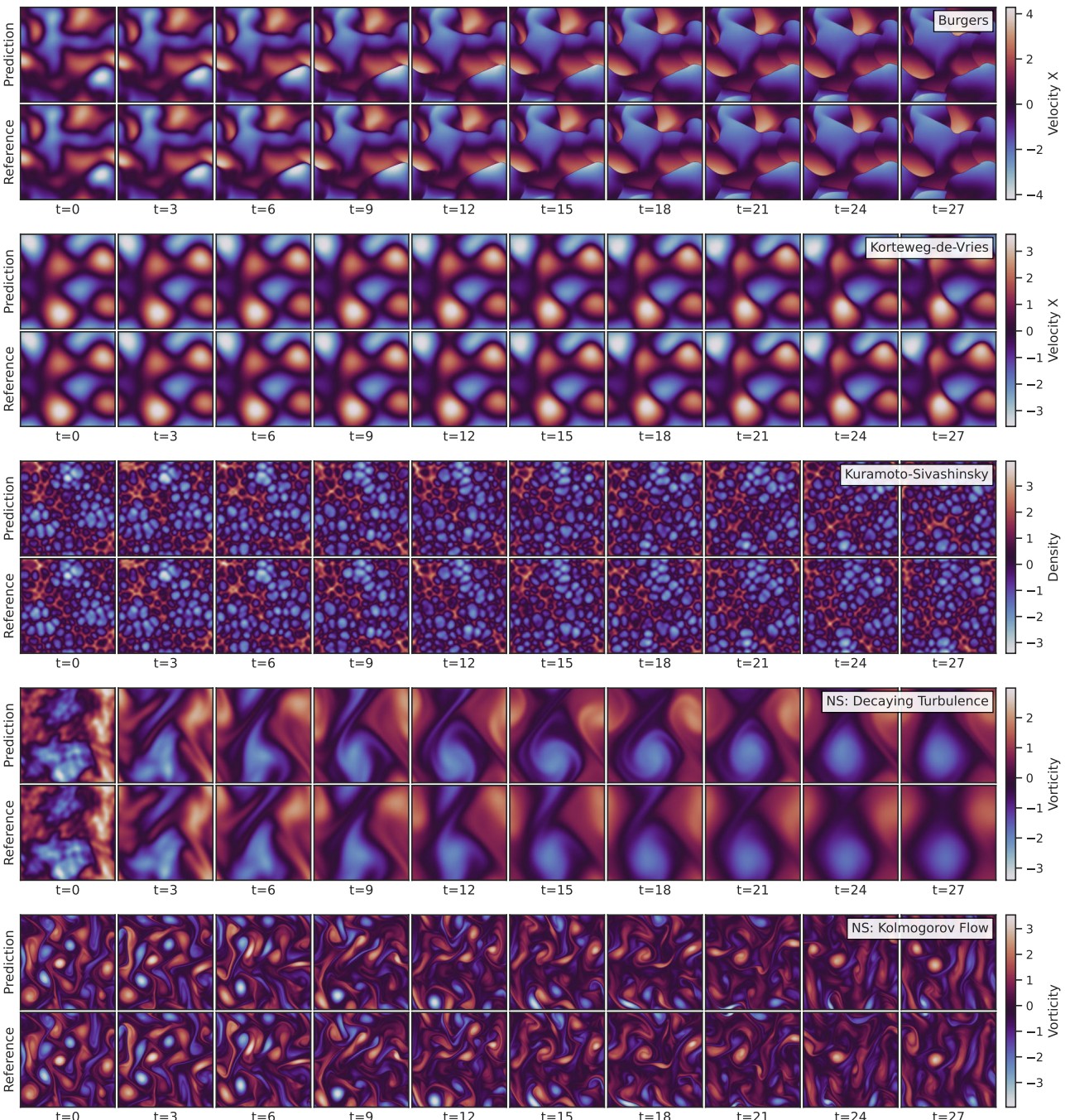

*Figure 23.* Visualizations of model's prediction on `burgers`, `kdv`, `ks`, `decay-turb` and `kolm-flow`.

# E. Downstream tasks

## E.1. Dataset

We introduce three challenging PDE prediction missions, which are active matter, Rayleigh-Bénard convection, and shear flow, as the downstream tasks. Fig. 24 shows the visual examples of these three tasks. All the simulations are from the *Well* dataset (Ohana et al., 2024). For each provided dataset, a predefined data split in training, validation, and test set already exists in the *Well* dataset. We randomly select 42, 8, and 10 trajectories from the corresponding split data for training, validation, and testing, respectively. Each trajectory is also truncated randomly to 30 frames. Details for each dataset are discussed as follows:

**Active Matter**   features simulations of rod-like biological active particles immersed in a Stokes flow. The active particles transfer chemical energy into mechanical work, leading to stresses that are communicated across the system. Furthermore, particle coordination causes complex behavior inside the flow. The following overview summarizes key characteristics of the dataset (for further details see Maddu et al., 2024):

- Boundary Conditions: periodic
- Time Step of Stored Data: 0.25
- Spatial Domain Size of Simulation: $[0, 10] \times [0, 10]$
- Spatial Resolution: $x = 256, y = 256$.
- Fields: concentration, velocity $(x, y)$. The orientation $(xx, xy, yx, yy)$ and strain $(xx, xy, yx, yy)$ fields in the original *Well* dataset are dropped in the current test.

**Rayleigh-Bénard Convection**   contains simulations of Rayleigh-Bénard convection on a horizontally periodic domain. It combines fluid dynamics and thermodynamics, by simulating convection cells forming due to temperature differences between an upper and lower plate. The combination of buoyancy, conduction, and viscosity leads to complex fluid behavior with boundary layers and vortices. The following overview summarizes key characteristics of the dataset (for further details see Burns et al., 2020):

- Boundary Conditions: periodic $(x)$, wall $(y)$
- Time Step of Stored Data: 0.25
- Spatial Domain Size of Simulation: $[0, 4] \times [0, 1]$
- Spatial Resolution: $x = 512, y = 128$.
- Fields: buoyancy, pressure, velocity $(x, y)$

**Shear Flow**   features simulations of the periodic incompressible Navier-Stokes equations in a shear flow configuration, where two fluid layers are sliding past each other at different velocities. Predicting the resulting vortices across different Reynolds and Schmidt numbers has important automotive, biomedical, and aerodynamics applications. The following overview summarizes key characteristics of the dataset (for further details see Burns et al., 2020):

- Boundary Conditions: periodic
- Time Step of Stored Data: 0.1
- Spatial Domain Size of Simulation: $[0, 1] \times [0, 2]$
- Spatial Resolution: $x = 512, y = 256$.
- Fields: density, pressure, velocity $(x, y)$

## E.2. Autoregressive Predictions

We show autoregressive predictions with the pre-trained PDE-S and separate channels on the training set for each dataset in Figures 25 to 27.

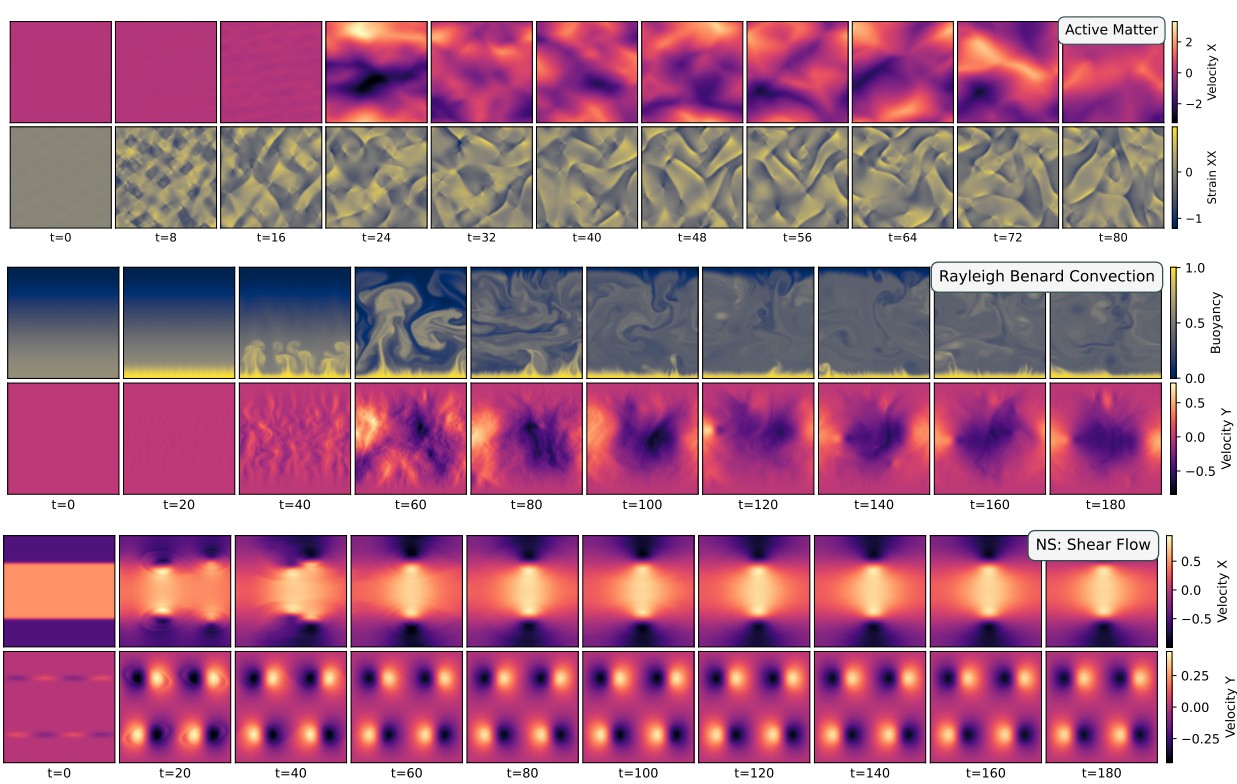

*Figure 24.* Random example simulations from active matter, Rayleigh-Bénard convection, and shear flow.

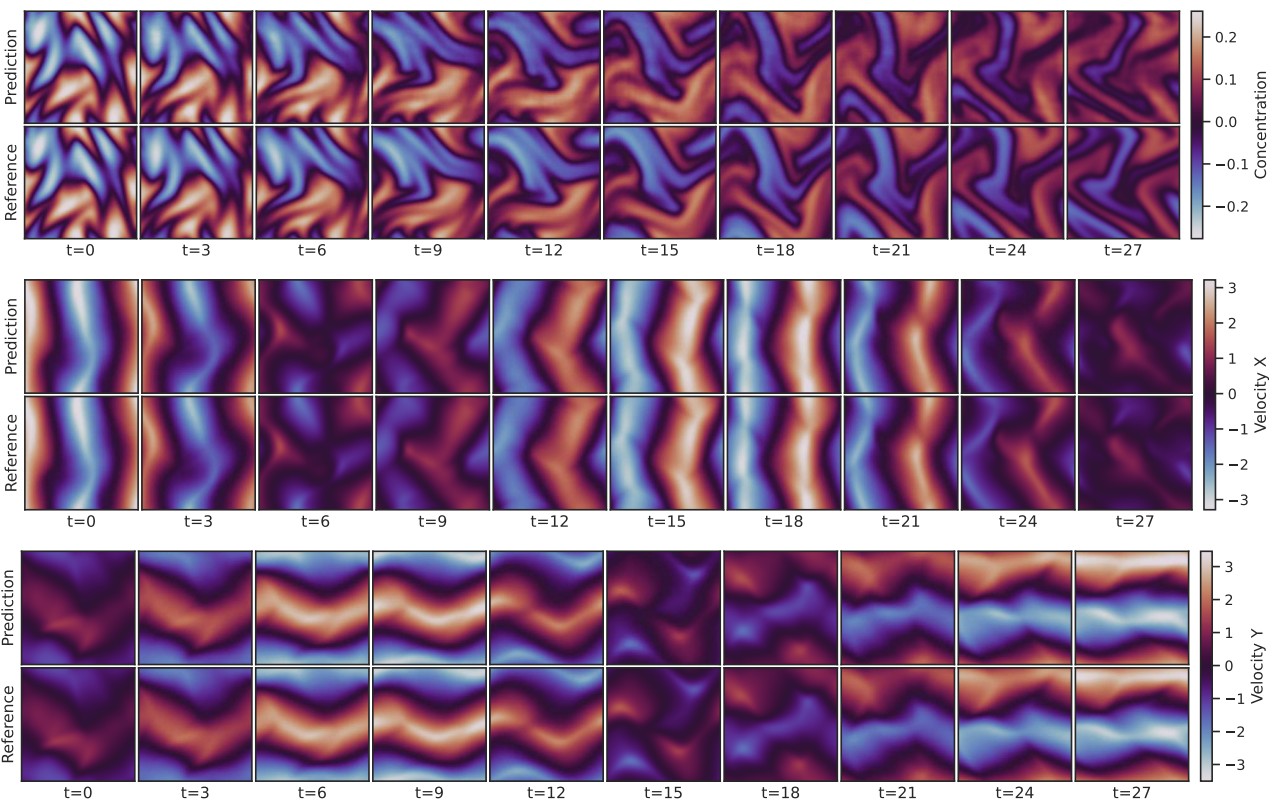

*Figure 25.* Active Matter. Autoregressive prediction with pre-trained PDE-S (SC).

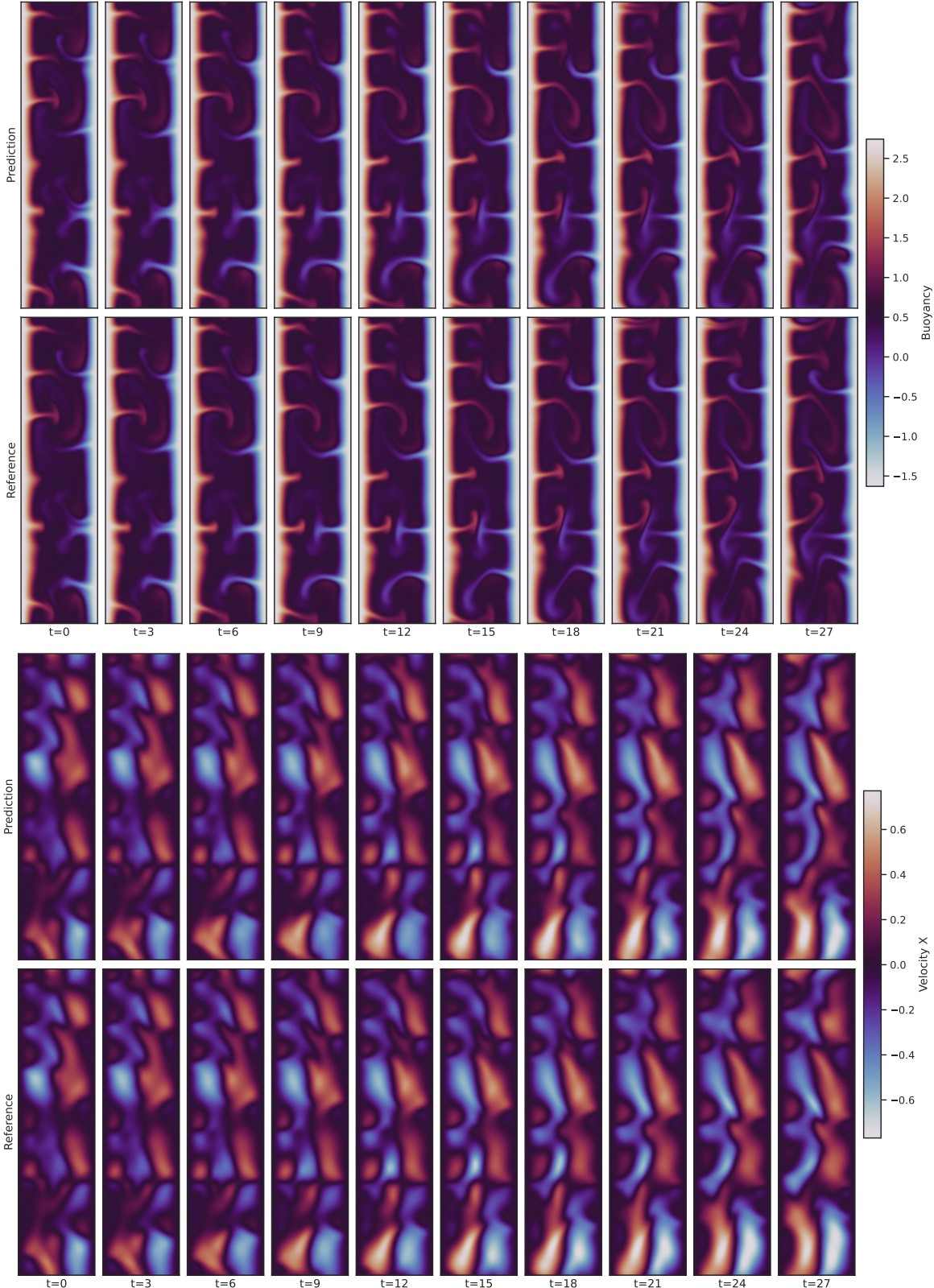

*Figure 26.* Rayleigh-Bénard Convection. Autoregressive prediction with pre-trained PDE-S (SC).

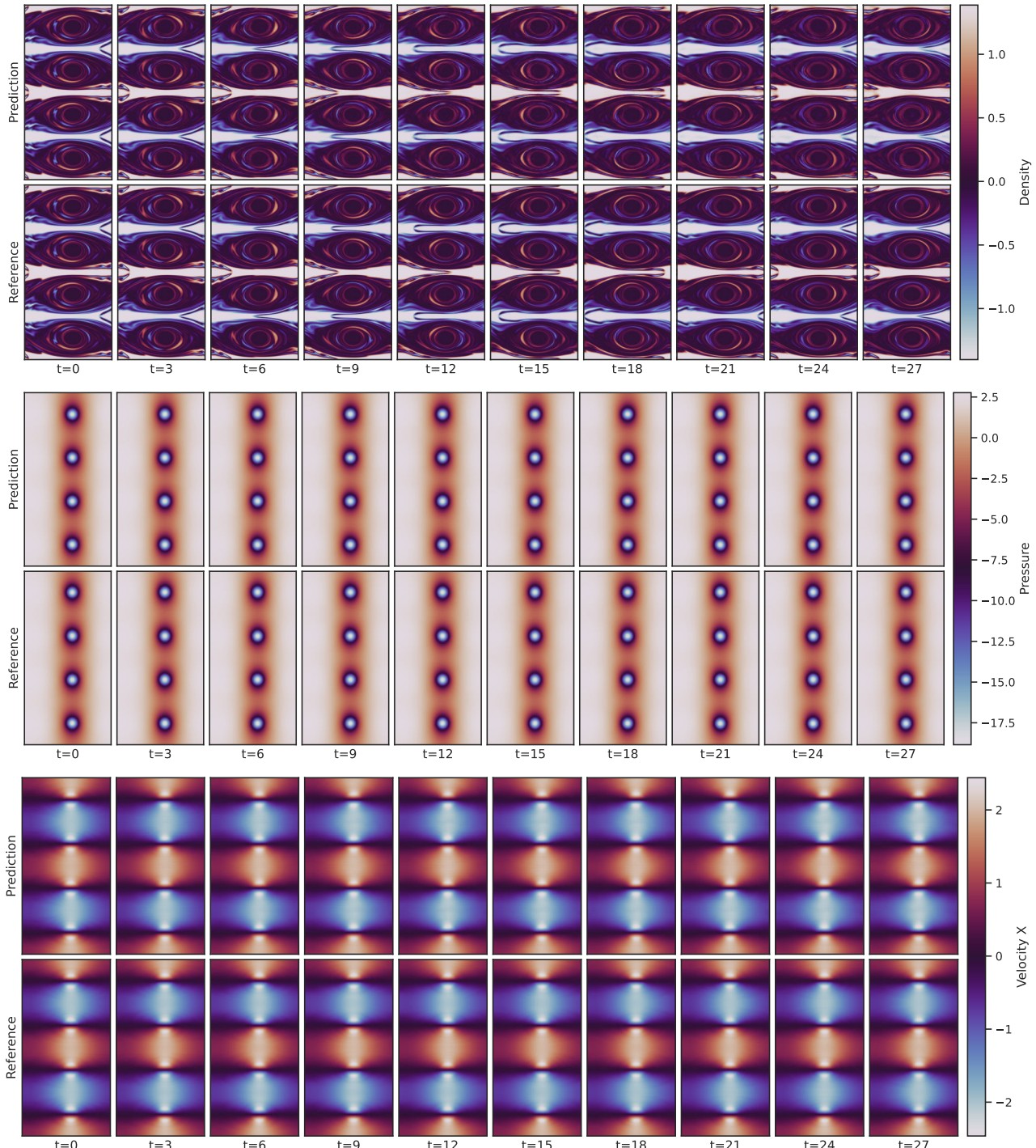

*Figure 27.* Shear Flow. Autoregressive prediction with pre-trained PDE-S (SC).

