# OpenReview forum: "PDE-Transformer: Efficient and Versatile Transformers for Physics Simulations"
_ICML.cc/2025/Conference — ICML 2025 poster_

### Official Review · Reviewer_yvZY · 2025-03-06

**Overall Recommendation:** 1

**Summary:**

This paper presents an enhanced diffusion transformer architecture through the integration of several established techniques, including multi-scale modeling and shifted window attention mechanisms. The improved model is subsequently applied to partial differential equation (PDE) solving tasks. Extensive experimental evaluations demonstrate the effectiveness of the proposed architectural modifications.

**Claims And Evidence:**

All claims are clear for me.

**Essential References Not Discussed:**

It is worth noting that UniSolver [1], a recent work also built upon the diffusion transformer architecture, addresses similar PDE generalization challenges. A detailed comparison and discussion with UniSolver should be included to highlight the distinctions and relative advantages of the proposed approach.

[1] Unisolver: PDE-Conditional Transformers Are Universal PDE Solvers

**Experimental Designs Or Analyses:**

The experiments and analyses are extensive and content-rich. The writing is clear and well-structured.

**Methods And Evaluation Criteria:**

The proposed method appears to primarily incorporate well-established techniques in deep learning, making the specific contributions less distinct. Additionally, the experimental evaluation employs relatively weak baselines. It would be beneficial to include comparisons with recent advancements in PDE foundation models and neural PDE solvers [1,2] for more comprehensive assessment.

[1] Scalable Transformer for PDE Surrogate Modeling

[2] PDEformer: Towards a Foundation Model for One-Dimensional Partial Differential Equations

**Other Comments Or Suggestions:**

N/A

**Other Strengths And Weaknesses:**

The main concern is lack of novelty.

**Questions For Authors:**

N/A

**Relation To Broader Scientific Literature:**

N/A

**Theoretical Claims:**

This paper does not have any theoretical claim.

---

> ### Author Rebuttal · Authors · 2025-03-31
>
> Thank you for the review and feedback. We want to address your remaining concerns in the following:
>
> **Stronger Baselines** While there can certainly be more baselines, we politely disagree that the baselines chosen are "relatively weak". For example, we compare extensively against scalable operator transformer scOT [1a] and UDiTs [2b]. They are both very strong and recent baselines (both published NeurIPS) for scientific machine learning and a SOTA diffusion transformer architectures.
>
> Thank you for mentioning the paper [3c], FactFormer, which is also a possible baseline, so we include it in our comparison of transformer architectures for a more comprehensive evaluation. See the following table for a comparison between FactFormer and PDE-Transformer (extending table 1 in the main paper):
>
> | Model    | nRMSE(1) | nRMSE(10) | Time (h) | Params | GFlops |
> | -------- | ------- | ------- | ------- | ------- | ------- |
> | PDE-S  | 0.044  | 0.36 | 7h 42m | 33.2M | 19.62 |
> | FactFormer | 0.069 | 0.65 | 38h 8m | 3.8M | 66.76 |
>
> PDE-S clearly outperforms FactFormer. Note that FactFormer has fewer parameters, since we were using the implementation by the authors of [3c] (from github). As a weight-computation tradeoff, we preserved the original architecture with fewer weights, but gave FactFormer a significantly larger computational budget. Hence, despite fewer parameters than PDE-S, FactFormer trains much longer and requires more than three times as many floating point operations. We believe this is fair comparison for FactFormer. The resulting performance of FactFormer is significantly lower despite the additional operations.
>
> For the PDEformer model [4d], we will include it in the related work. PDEformer only targets 1D PDEs and an extension to 2D PDEs seems nontrivial. Moreover, it constructs a graph using the target PDE, which is not always available in the general setup we are targeting. Similarly, Unisolver [5e] is mostly orthogonal to our work: it primarily focuses on conditioning on PDE specific information (PDE equation, boundaries, etc.) using language embeddings from LLMs. Note that both [4d] and [5e] are only available as preprints so far.
>
>
> [1a] Poseidon: Efficient Foundation Models for PDEs, https://arxiv.org/pdf/2405.19101
>
> [2b] U-DiTs: Downsample Tokens in U-Shaped Diffusion Transformers, https://arxiv.org/pdf/2405.02730v1
>
> [3c] Scalable Transformer for PDE Surrogate Modeling, https://arxiv.org/pdf/2305.17560
>
> [4d] PDEformer: Towards a Foundation Model for One-Dimensional Partial Differential Equations, https://arxiv.org/pdf/2402.12652
>
> [5e] Unisolver: Unisolver: PDE-Conditional Transformers Are Universal PDE Solvers, https://arxiv.org/pdf/2405.17527
>
> **Novelty** Even though PDE-Transformer combines improvements from different more established architectures in computer vision, the final architecture is novel and follows the paradigm “use what works best”. It gives SOTA performance on learning PDEs with significantly improved scalability. The modifications are carefully evaluated against the newest, similar SOTA transformer architecture. Additionally, the separate channel (SC) variant is not used in any previous work, and fundamentally improves the downstream performance. For building scientific foundation models, this is one of the most critical aspect: effective pretraining on large datasets so that finetuning on difficult new PDEs works. We have shown that finetuning a pretrained network works much better for the separate channel SC version than when mixing channels (MC). We believe this is an important empirical finding.
>
> We therefore kindly ask you to reconsider your overall recommendation, and we’d be happy to discuss any remaining open aspects.

---

### Official Review · Reviewer_2XVZ · 2025-03-10

**Overall Recommendation:** 3

**Summary:**

The paper presents a transformer model called PDE Transformer designed to solve PDE (partial differential equations), therefore potentially allowing for physical simulations. The model is based on the diffusion transformer architecture (DiT), and as such, can be be trained not only for forecasting, but also for generation. To the best of my understanding, one of the main additions to the architecture is to use down- and upscaling tokens at the end of each transformer stage (instead of on the query- key-value tuple of the self-attention operation like it was introduced before in another version of the architecture) which allows for a much faster training time, and the application of the architecture to a PDE learning setting. The paper claims to beat SOTA architectures on a benchmark PDE forecasting dataset including multiple PDEs, and for so called “downstream tasks”: more complex PDEs with various boundary conditions on specific geometries.

# Update after the rebuttal

I appreciate the efforts made by the authors during the rebuttal and I think the paper is of overall good quality. I am maintaining my score as it is.

**Claims And Evidence:**

The claims are clear: the suggested model outperforms SOTA models for deep learning based PDE simulation.

The evidence given is proper but not entirely convincing, considering the following points:
- The number of SOTA models it is compared to for the main PDEs is quite limited: 3 models, only transformer based. Why not comparing to other types of PDE-learning models that are also quite efficient and always competing such as graph based models and neural operator based models (e.g. Message passing neural pde solvers from Brandstetter et al 202, MagNet boussif et al 2022, or FNOs, which I don't need to introduce) ?
- I am honestly not sure why different SOTA models are used for the main PDE prediction tasks (trained from scratch) and the “downstream” tasks? They can all do both predictions, so we might as well see the performances of all models on all tasks!
- Also, one of them (UDiT) is basically performing equally to PDE-Transformer, although it is slower at training time. The suggested model is therefore indeed faster, but not necessarily “outperforming” SOTA. Perhaps this should be highlighted in the main claims!
- The configurations in which the model is tested is also quite limited: only one spatial resolution is used (and quite a corse one for typical PDE datasets: 256x256) and one time resolution (30 steps for the 600 different PDE trajectories) for a rather low horizon of 10 steps. I appreciated the studies on the patch and window sizes and supervision vs probabilistic learning, but I think time and space resolutions are of paramount importance for PDE learning (if I had to choose, I would put extra studies in the appendix and these on resolution in the main paper if it causes a space problem). Some models might be more efficient for longer horizon and/or higher spatial resolution! I think it would be more fair to do more experiments regarding various resolutions and conclude on the good and bad points of multiple models comparatively (For example FNO and variants are known to be quite good at zero-shot super resolution in both space and time tasks)
- The same two previous comments can be said for the “downstream tasks”: 4 models are compared, including some models that are not adapted to strange geometries (which is mentioned by the authors, which I appreciate!). Similarly experiences with various resolutions would be interesting.
- Note that I really appreciated the pretrained experiments on the downstream tasks, I think it is interesting and impressive; however, I did not understand if the other models were also retrained or only trained from scratch? If the latter, the comparison with the pretrained PDE-transfo is a little unfair. (Cf questions for more details)

**Essential References Not Discussed:**

Relating to the SOTA models previously mentioned in the review, it appears that the authors do not mention PDE learning models based on graph neural networks, while it was of very high significance in the domain at some point, particularly for their super resolution and irregular mesh learning capabilities (and I believe they still are). It is of course necessary to compare to recent transformer models, but at the end of the day all significant SOTA models should be considered, regardless of the architecture type.

Such graph based models include for example:
Message passing neural PDE solvers: Brandstetter, J., Worrall, D., & Welling, M. (2022). Message passing neural PDE solvers. arXiv preprint arXiv:2202.03376.

MAgNet: Boussif, O., Bengio, Y., Benabbou, L., & Assouline, D. (2022). MAgnet: Mesh agnostic neural PDE solver. Advances in Neural Information Processing Systems, 35, 31972-31985.

I am not saying you need to compare to all kinds of models that exist, that would be impossible! But do believe the SOTA models used in comparison needs to include some of these graph based models, in terms of pure forecasting performance, but also super resolution abilities and training time (since it is one of the biggest advantage of the model).

**Experimental Designs Or Analyses:**

The experimental design seems perfectly sound, except that, as mentioned in the Claims And Evidence section, more experiments regarding spatial and temporal resolutions, higher time horizons and more SOTA models would bring more convincing evidence to the suggested model’s performance.

**Methods And Evaluation Criteria:**

The proposed methods do make sense, as it basically builds on top of successful architectures, which may not be the most creative but makes total sense! The benchmark dataset is good and gathers many PDEs (APEBench). Perhaps it would make sense to show the typical multiple metrics for forecasting (MAE, MSE etc rather than only nRMSE ) in order to have a full view on the models performances. Also, more experiments would be needed to have a definite conclusion, as mentioned previously.

**Other Comments Or Suggestions:**

- Is the idea of “expansion rate” really useful? It seems to me that it only illustrates the tradeoff between better performance/scalability obtained with lower/higher patch(token) size, which is already known! Also, you are mentioning it in the abstract but you do not define it there but only later in the article. If you want to keep presenting this notion, please define it in the abstract before mentioning it or dont mention it!
- When the nRMSE is defined, a dependency to horizon time could be added (meaning defining it as nRMSE(t) ) since it is presented like that in table 1, to be more coherent with the notations.

**Other Strengths And Weaknesses:**

Strengths:
- The paper is well written and well presented, and rather clearly explained.
- I appreciate
    - the practical idea of using up and downsample tokens in between transformer stages instead of in attention computation to make it more time efficient and
    - The idea of using a diffusion model to the PDE learning domain (as it seems to work pretty well!)

Weaknesses:

The main weakness is therefore the originality of the paper, given than its contribution is mainly residing in a very slight practical change in an already existing architecture and the application of the architecture to PDE learning. This could be interpreted as a “real-world” use case since PDE learning can be directly applied for physical simulations, however the paper does not really take it this far, by doing actual physical simulations (combining potentially multiple PDEs) and rather tries it directly on PDE data.
I do believe it is still a useful contribution, but then I think it would greatly benefit from stronger experiments (on multiple spatial and time resolution experiments, as mentioned previously) to counter balance the lack of contributions in other aspects.

**Questions For Authors:**

- Note that I really appreciated the pretrained experiments on the downstream tasks, I think it is interesting and impressive; however, I did not understand if the other models were also retrained or only trained from scratch? If the latter, the comparison with pretrained PDE-transo is a little unfair, and it would be interesting to see the performance of these models pre-trained as well in comparison.
- In addition to additional experiments, I think it would be interesting to consider super resolution experiments, in space and/or time, as I think it is very important in PDE learning (the ability to have a higher resolution for the prediction, based on coarser resolution models like global weather forecasting models, is more important to me than a faster training time - once a model is trained in real life, what matters is rather the inference time), and since one of the bigger contribution is basically the application of a known architecture in PDEs, it would be significant  to see if it compares to neural operators models and graph neural networks models which are both adapted to these tasks (using respectively implicit functions and mesh agnostic nodes)

**Relation To Broader Scientific Literature:**

The contribution of the paper seems to be in terms of pure performance in the PDE learning domain, as it competes with state of the art while being apparently faster to train than some SOTA PDE models. In terms of pure architecture, it is however not very original, as it is basically the UDiT architecture but with up and downsample tokens added at the end of transformer stages instead of the attention dot products themselves, applied to a PDE forecasting task.
It can be seen from the performance table that the only advantage of the PDE transformer over UDiT is the training time, but the performance is extremely close.

**Theoretical Claims:**

No theoretical claims are given in this paper.

---

> ### Author Rebuttal · Authors · 2025-03-31
>
> Thank you for the positive review and feedback.
>
> **Pretraining on Downstream Tasks** Baseline models are not pretrained. We have a version of our own model that is trained from scratch and one that is pretrained. In all cases, our model trained from scratch is better than the baselines and performance is further improved if it is pretrained, so we think that's fair for a comparison. We also compare with Poseidon (scOT-B, see figure 9 in the appendix) which is pretrained by the corresponding authors on their own large pretraining dataset.
>
> **Superresolution experiments** In principle, PDE-Transformer can be trained on a coarse resolution and then applied to high-resolution data later. This is tied to both how the attention operation can be viewed as an integral kernel operation (similar to many neural operator papers). Within each attention window, a 2D grid with relative positions is used as an input to a MLP that outputs position-dependent attention scores. We can modify the relative position grid and window size to obtain a model that operates at an increased resolution.
>
> Because of two central modifications that improve the scalability of the architectures, super-resolution without finetuning is difficult to achieve. Those two modifications are patching and the down- and upsampling of tokens within the multi-scale architecture, which both assume data to be at a fixed resolution.
>
> Based on your suggestions, we have evaluated PDE-Transformer trained at resolution 256^2, modified the window size and evaluated the modified model at resolution 512^2. While results still showed the correct dynamics, the accuracy was affected. When finetuning PDE-Transformers to new resolutions we expect the proposed architecture to generalize well and achieve a high performance.
>
> **Graph-based models**
> We agree that graph-based models are an important baseline for unstructured meshes. For regular grids, previous work has shown that graph networks have no advantages, but only require a slightly larger weight count compared to networks that leverage the inductive biases from the grid. See e.g. [2] for a comparison.
>
> Nonetheless, we have started to train Message Passing Neural PDE Solvers (MPNN) on the pretraining dataset and plan to add a comparison in the updated manuscript. In the MPNN paper, the 2D experiments only involved grids of size 32^2. For the 256^2 resolution, training has started, but takes several days to finish, because of much bigger memory and compute requirements. Analyzing the architecture, a clear conclusion to draw is that graph networks like MPNN are not designed to be scaled up to large-scale tasks. Due to the required training time, we can only report very preliminary results (epoch 2/100) for the rebuttal.
>
> | Model    | Epoch | nRMSE(1) | nRMSE(10) | Time (h) | Params | GFlops |
> | -------- | ------- | ------- | ------- | ------- | ------- | ------- |
> | PDE-S  | 100 | 0.044  | 0.36 | 7h 42m | 33.2M | 19.62 |
> | MPNN | 2 | 0.283 | 1.07 | 37h 2m (1 GPU) | 2.44M | 396.16 |
>
> As mentioned above, we plan to target unstructured grids and datasets as future work, and will make sure to compare to graph networks more extensively for these cases.
>
> [2] Differentiability in Unrolled Training of Neural Physics Simulators on Transient Dynamics, https://arxiv.org/pdf/2402.12971
>
> **Different models for pretraining/downstream tasks**: We considered different models for pretraining and the downstream tasks. First, we wanted to focus on transformer architectures that are more similar to our architecture for pretraining. For the downstream tasks, we have more "standard" scientific ML baselines. It would be possible to train every model on all tasks, giving an even more extensive comparison, but we believe that grouping models (transformer/scientific ML) is a fair approach that demonstrates relative performance differences. There is one architecture (scOT) that fits in both categories and it was included in both pretraining/finetuning tasks. We included an additional SOTA transformer model, FactFormer. See our response to reviewer yvZY.
>
> **UDiT-S better than PDE-S?** In Table 1, UDiT-S is *slightly* better than PDE-S for nRMSE(1). Still, in almost any case it is better to use PDE-S. The different configs S,B,L are not always directly comparable between model architectures, e.g. UDiT-S has more parameters than PDE-S. The performance of transformer models is known to increase when scaling the model size, so a "larger" UDiT-S can beat a "smaller" PDE-S. The B config PDE-B trains much faster than UDiT-S (10h40m vs. 18h 30m) but now clearly beats UDiT-S in terms of nRMSE (0.038 vs. 0.042). Inference speed is correlated to training time. UDiT does not scale well for higher resolutions as demonstrated in figure 3.
>
> We will improve the paper based on your many other helpful suggestions. We had to keep our answers short due to the strict rebuttal character limit.
>
> We are happy to continue in more detail during the discussion phase.

---

### Official Review · Reviewer_aHgi · 2025-03-12

**Overall Recommendation:** 3

**Summary:**

The paper presents PDE-transformer, a new transformer approach trained simultaneously on different physical systems. They propose an alternative to the DiT architecture that is suited for PDEs. Specifically, they use a multi-scale architecture and a shifted-window attention to prevent from the quadratic complexity. They propose also a separate channel strategy when dealing with physical systems with different number of physical channels. They evaluate their method on a variety of PDEs with specific behaviors and complex PDEs for finetuning, where they show competitive or sota performance.

## update after rebuttal
The authors proposed a nice approach for multi-task PDE solving. There are some results that are surprising to me, that have been mentioned in the rebuttal. Overall, the paper is of good quality and thus I keep my score as it is.

**Claims And Evidence:**

Yes

**Essential References Not Discussed:**

The authors discussed the main related works

**Experimental Designs Or Analyses:**

I did check the different experimental designs and analyses and did not find a particular issue. However, I am a bit concerned by the analysis made on the diffusion training. They are known to correctly model complex distributions, which could be the case when pretraining a model on a variety of PDEs.

**Methods And Evaluation Criteria:**

They used a variety of datasets for their experiments and also tested on very hard datasets  for some additional studies, showing that using a pretrained transformer on more simple datasets can help for downstream tasks such as finetuning on very complex PDEs.

**Other Comments Or Suggestions:**

No more comments. See above sections.

**Other Strengths And Weaknesses:**

The paper tackles an important problem, it learns a single model for solving multiple physical systems. It exhibits strong performance on a diverse number of datasets and have been tested on very complex datasets in a downstream task, showing strong performance.

The paper is clear and I particularly liked the appendix with all the dataset descriptions.

The originality of the paper is a bit weak, it extends existing works from computer vision with DiTs for scientific problems. Using U-shape transformers is not novel, but it's application for PDEs with a multi-scale approach makes sense.

One potential weakness of the paper is the restriction to flexible grids, which should be a key property of neural solvers. I think it could be easily extended to handle irregular grids.

**Questions For Authors:**

I am bit concerned by the conclusion made with MSE training compared to diffusion training. I don't really see why MSE training could yield superior performance, notably when modeling complex distributions such as multi-physics data. In my opinion, diffusion training should exhibit superior performance. Could you elaborate more on why MSE training is superior here please.

**Relation To Broader Scientific Literature:**

The authors proposed a model in the line of the existing foundation models for PDEs literature.

**Theoretical Claims:**

No theoretical claims

---

> ### Author Rebuttal · Authors · 2025-03-31
>
> Thank you for the positive review and feedback.
>
> **MSE vs. Diffusion** This is a very important and interesting issue. We believe that scientific machine learning models should in general be capable of learning the full posterior; however how useful this is still depends on the specific task and the underlying data distribution.
>
> When training a neural solver for example, then the mapping that the neural solver learns is deterministic when given simulation hyperparameters and initial conditions. Another way to phrase this is that given a point in the input space (representing the simulation hyperparameters and initial conditions), then this is mapped to a single point in the output space (the solution field at a certain time).
> In this situation, supervised training is a suitable choice, especially if we use a metric based on the MSE/L2 distance for evaluation and training.
>
> This situation changes, when the mapping becomes probabilistic, for example when the exact simulation hyperparameters are unknown. This case more closely resembles the setup in the paper. Then, instead of learning a deterministic mapping, we aim to learn a mapping that transports a single point in the input space to a distribution of possible solutions in the output space.
> In this case, we can learn the mapping via a diffusion model. What happens when we train a diffusion model, but use the common MSE/L2 distance for evaluation? For simplicity, let's say that this distribution is approximately Gaussian. We only have *samples* from the target distribution that we can compare to. At the same time, our diffusion model will draw a sample from the learned target distribution, which will match the target distribution if our model is trained well.
>
> Now, we evaluate the MSE/L2 distance between the sample from the target distribution and the sample from the "learned" distribution. Let's denote this value by the random variable X. One factor plays a key role now: if we use a metric based on MSE/L2 distance, then it is not optimal when the diffusion model learns the full posterior. In fact, it would be better, if the model just learned to predict the *mean* of the posterior, because that will decrease the mean and variance of X. If we use supervised training, the network will learn exactly the mean of the posterior.
>
> To summarize, if we use metrics based on MSE/L2 distance for the evaluation, then in theory they will favor supervised models over diffusion models, because it is just not optimal and incentivized to learn the full posterior for these metrics.
>
> Of course that does not mean that the supervised models are better than diffusion models. In fact, we advocate for using diffusion models. However, if the main objective is based on MSE, then it is very difficult to beat a supervised model in terms of the accuracy-compute tradeoff. For more complicated data domains and tasks, we should always use expert knowledge and metrics that make sure the full posterior is considered. Then we can reasonably show the advantages of diffusion models over supervised training.
> In this paper, we use nRMSE, because of the large number of different PDEs and because there is no single other metric that would work well for all PDEs. For nRMSE, we see that the diffusion models improve significantly when increasing the number of parameters from config S to L and when increasing the number of steps used for inference, closely approaching the supervised baseline.
>
> **Generalization to Unstructured Meshes**
>
> It's possible to generalize the the architecture to irregular grids, which however this is not the focus of the current paper.
>
> - One straightforward approach is to couple PDE-Transformer with GNO layers as encoder/decoders (instead of the patchification) that map from a given geometry to a latent regular grid as used in e.g. [1]
>
> - Alternatively, it is also possible to generalize the notion of attention window to be defined on local neighbourhoods. Correspondingly, token up- and downsampling are replaced by respective graph pooling operations. This requires more fundamental changes to the architecture making it a type of graph neural network.
>
> Both approaches are planned as directions of future work.
>
> [1] Geometry-Informed Neural Operator for Large-Scale 3D PDEs, https://arxiv.org/pdf/2309.00583
>
> Please let us know if this could clear your open questions. We are happy to discuss more.

---

### Official Review · Reviewer_hdWy · 2025-03-12

**Overall Recommendation:** 3

**Summary:**

This paper introduce PDE-Transformer, a transformer made for PDE data being able to be incorporated in a supervised learning task or in a diffusion model. The model is extensively tests on various benchmarks.

**Claims And Evidence:**

Yes, the claims seem supported by convincing evidence

**Essential References Not Discussed:**

None that I can think of.

**Experimental Designs Or Analyses:**

The experiments are extensive and look good to me.

**Methods And Evaluation Criteria:**

The authors benchmark their models on many datasets and study the transfer properties to datasets unseen during training.

**Other Comments Or Suggestions:**

- Line 158 (right paragraph) should be $S$ is and not S.
- Line 210 (r.p.) should be separate (typo)

**Other Strengths And Weaknesses:**

Strengths:
- The paper is very dense with many different ideas. All these ideas are supported with numerical experiments.
- The results are convincing.
- I like the downstream tasks study.

Weaknesses:
- The model is currently limited to 2D regular grids
- The paper is very dense and makes the architecture not 100% clear in 1 read.
- The fact that the model can be use in supervised and diffusion training gives me mixed feelings: it is a good idea, but overall, what does a diffusion models additionally bring here? It doesn't seem to be discussed.
- From the conclusions of Table 3, it is not clear to me which one of MC or SC is better.

**Questions For Authors:**

- Can the authors comment on the use of bf16 precision? This seems like a bold choice to me for this type of data which is usually in fp32. With bf16, don't you lose a lot of precision, especially for autoregressive tasks?
- One clear advantage of using diffusion models here would be that we would obtain probabilistic samples. It could lead to uncertainties in the prediction, very valuable for scientific machine learning. Can the authors comment on that, and possibly suggest 1 experiment to study this (not mandatory, but I think it would strengthen the paper)?
- There are some unclear terms in the text, what is token upsampling/downsampling? PixelShuffle/unshuffle?
- Are images just patchified? No MLP/convolutions? Do the authors know whether their better performance is because they are working in pixel space?
- Why did the author not compared with other foundation models like MPP and Poseidon?

If the weaknesses and questions are satisfyingly answered, I am willing to modify my score accordingly.

**Relation To Broader Scientific Literature:**

The contributions of the paper are a good progress towards better architectures in the field.

**Theoretical Claims:**

No theoretical claims.

---

> ### Author Rebuttal · Authors · 2025-03-31
>
> Thank you for the positive review and feedback.
>
> **BF16 mixed precision** That's a good point. With BF16 mixed precision we lose precision compared to FP32, but can train a lot faster. In our experiments, we did not see a difference in the evaluation metrics and training loss when switching between BF16 mixed precision and FP32. Note that in practice we can always train first with the faster BF16 mixed precision until the training loss converges and then finetune with full FP32 precision.
>
> **Diffusion models** In our opinion, scientific foundation models should be designed with the possibility in mind to produce probabilistic samples. Uncertainty estimates are especially useful when dealing with turbulence. Direct numerical simulation is very expensive, so more simplified turbulence models such as Reynolds-averaged Navier-Stokes simulation  and large eddy simulations are still prevalent in the engineering community. However, these models produce posterior distributions, because of unknown latent parameters of the models. As an example of an application that requires the full posterior, see [1] which learns the supersonic flow around aircraft airfoils using diffusion models.
> Even more computationally challenging is turbulence in 3D, which can be phrased as a generative modeling task to learn all possible turbulent flow states [2]. Scaling 2D models to 3D requires very computationally efficient 2D models to begin with, which we focus on in this paper. We are working on our follow-up work, which addresses efficient scaling of our transformer architectures to high resolutions in 3D.
>
> [1] Uncertainty-aware Surrogate Models for Airfoil Flow Simulations with Denoising Diffusion Probabilistic Models, https://arxiv.org/pdf/2312.05320v1
>
> [2] From Zero To Turbulence: Generative Modeling for 3D Flow Simulation, https://arxiv.org/pdf/2306.01776
>
> **Token Upsampling/Pixel Shuffle** Token downsampling refers to an operation that merges multiple (4) tokens into a single token. Token downsampling can be implemented for example via PixelUnshuffle, which is part of the Pytorch library (see torch.nn.PixelUnshuffle of the official Pytorch documentation). Respectively, there is token upsampling, which splits a single token into multiple (4) tokens, which can be implemented with PixelShuffle.
>
> **Patchification** Our implementation uses a single convolutional layer for the patchification where the kernel size and stride are set to the patch size, which is an efficient standard implementation of the patchification. We have performed experiments with extending this single convolutional layer for patchification to small convolutional encoders/decoders.
> However we didn't see any improvements here and using a pure transformer backbone with a single patchification layer was optimal for us.
> Our transformer also works in pixel space directly. In principle it can be easily coupled with additional encoders/decoders. However, doing this needs to be carefully engineered. In computer vision, transformer architectures such as the Diffusion Transformer, are mostly coupled with pretrained VAEs and work in a reduced latent space. For scientific machine learning, a pretrained VAE is often problematic, as it makes it difficult to achieve low MSE values.
>
> **Which is better: MC or SC?**
> For the pretraining tasks, we are in a situation where there are very large amounts of simulation data to train on. In this case, the mixed channel (MC) and separate channel (SC) variants achieve the same accuracy, however SC requires more computation. In this case, MC still wins the accuracy-compute tradeoff. For finetuning, there are domain-specific applications, where data is more valuable and scarce. Here we see that when trained from scratch, MC generalizes better than SC. However, when we don't train from scratch and finetune pretrained networks, then (1) results always improve for both SC and MC and (2) SC now clearly beats MC and shows much bigger improvements from pretraining.
> This is because the SC version is better at transferring what it has learned to new types of simulations. Thus for typical use-cases of finetuning a pretrained network, SC is clearly preferable.
>
> **Comparison with Poseidon/MPP**
> We did in fact compare our model extensively to Poseidon. The model from Poseidon is the scOT model we compare to in all experiments. We also compare to the pretrained model, Poseidon-B for the downstream tasks. Because of space limitations, we could not include a full comparison in the main text, but it is discussed and we refer to figure 9 in the appendix.
>
> We saw Poseidon as the most recent and "toughest" competitor, so we wanted to make sure we have extensive comparisons with it and show that we can outperform it. MPP is already used as a baseline in the Poseidon paper and Poseidon achieves much better performance there, so our priority was to focus on Poseidon.
>
> Please let us know if this could clear your open questions. We are happy to discuss more.

---

### Decision · Program_Chairs · 2025-05-01

**Decision:**

Accept (poster)

**Comment:**

The authors introduce PDE-Transformer, an improved transformer-based architecture for modeling various physics simulations on regular grids. The method is limited to 2D regular grids. The method is demonstrated on a variety of PDEs, showcasing the generality and performance of the method. The reviewers are mixed. Three reviewers are in favor of accepting the work, while one objects to it. The three positive reviewers appreciate the paper’s quality, methodology, and added results. The fourth reviewer based some of the criticism on comparisons to very recent arxiv papers. Overall, based on the reviews, I recommend the acceptance of the paper to ICML.